# Microstructural changes precede depression in patients with relapsing-remitting Multiple Sclerosis

Frank Riemer [1,2✉], Ellen Skorve[2,3], Ofer Pasternak [4,5], Fulvio Zaccagna [6,7,8], Astri J. Lundervold [9], Øivind Torkildsen[2,3], Kjell-Morten Myhr[2,3] & Renate Grüner[1,10]

## Abstract

**Background** Multiple Sclerosis lesions in the brain and spinal cord can lead to different symptoms, including cognitive and mood changes. In this study we explore the temporal relationship between early microstructural changes in subcortical volumes and cognitive and emotional function in a longitudinal cohort study of patients with relapsing-remitting Multiple Sclerosis.

**Methods** In vivo imaging in forty-six patients with relapsing-remitting Multiple Sclerosis was performed annually over 3 years magnetic resonance imaging. Microstructural changes were estimated in subcortical structures using the free water fraction, a diffusion-based MRI metric. In parallel, patients were assessed with the Hospital Anxiety and Depression Scale amongst other tests. Predictive structural equation modeling was set up to further explore the relationship between imaging and the assessment scores. In a general linear model analysis, the cohort was split into patients with higher and lower depression scores.

**Results** Nearly all subcortical diffusion microstructure estimates at the baseline visit correlate with the depression score at the 2 years follow-up. The predictive nature of baseline free water estimates and depression subscores after 2 years are confirmed in the predictive structural equation modeling analysis with the thalamus showing the greatest effect size. The general linear model analysis shows patterns of MRI free water differences in the thalamus and amygdala/hippocampus area between participants with high and low depression score.

**Conclusions** Our data suggests a relationship between higher levels of free-water in the subcortical structures in an early stage of Multiple Sclerosis and depression symptoms at a later stage of the disease.

## Plain language summary

Signals between the brain and spinal cord are disrupted in people with Multiple Sclerosis. For those with relapsing-remitting Multiple Sclerosis (RRMS), symptoms get periodically better and worse over time. We looked at whether changes in the brain of people with RRMS were associated with changes in their mood over time. People who had more changes in certain areas of the brain at the start of the study were more likely to have symptoms of depression later. This work suggests that early changes in the brain may be linked to increased symptoms of depression over time in people with RRMS. We believe this could be an opportunity to provide care to those suffering from RRMS to lessen the impact of severe depression symptoms before they arise.

[1] Mohn Medical Imaging and Visualization Centre (MMIV), Department of Radiology, Haukeland University Hospital, 5021 Bergen, Norway. [2] Neuro-SysMed, Department of Neurology, Haukeland University Hospital, 5021 Bergen, Norway. [3] Department of Clinical Medicine, University of Bergen, 5020 Bergen, Norway. [4] Department of Psychiatry, Brigham and Women's Hospital, Harvard Medical School, Boston, MA 02215, USA. [5] Department of Radiology, Brigham and Women's Hospital, Harvard Medical School, Boston, MA 02215, USA. [6] Department of Imaging, Cambridge University Hospitals NHS Foundation Trust, Cambridge Biomedical Campus, CB2 0QQ Cambridge, United Kingdom. [7] Department of Radiology, University of Cambridge, CB2 0QQ Cambridge, United Kingdom. [8] Investigative Medicine Division, Radcliffe Department of Medicine, University of Oxford, OX3 9DU Oxford, United Kingdom. [9] Department of Biological and Medical Psychology, University of Bergen, 5020 Bergen, Norway. [10] Department of Physics and Technology, University of Bergen, 5007 Bergen, Norway. ✉email: f.riemer@web.de

Multiple Sclerosis (MS) is an inflammatory and degenerative central nervous system disease characterized by demyelination, axonal loss and gliosis[1]. Clinically, the disease is typically associated with impairment of motor and sensory function, but also cognitive and emotional functions that commonly lead to progressive disability with reduced quality of life and shortened life expectancy. The diagnosis is based on the history of symptoms, clinical findings, and measures derived from MRI examinations and cerebrospinal fluid (CSF) analyses[2]. Most patients (85–90%) experience a relapsing-remitting disease course (RRMS), and 10–15% experience a primary progressive (PPMS) or secondary progressive (SPMS) course. A hallmark of RRMS is high disease activity in terms of lesions with distinct attacks and remission periods, while PPMS and SPMS have fewer lesions and are marked by a gradual worsening of symptoms without any distinct attacks or remission periods[1]. New disease activity is most often visible as lesions on MRI images, and brain changes are expected to be present before the core clinical symptoms of the disease. In the present longitudinal study, we investigate associations between the brain measures and clinical symptoms in a group of recently diagnosed patients with RRMS.

Longitudinal recording of clinical relapses, disability progression evaluated by the Expanded Disability Status Scale (EDSS)[3], as well as the identification of new lesions on structural MRI images[2] are key elements when identifying and monitoring patients with MS. In addition to identifying lesions, MRI research has been employed to better understand the pathophysiology of the disease. Popular areas of MRI research aim to provide complimentary measures of inflammation[4], brain atrophy through volumetry[5] and using functional MRI (fMRI) experiments to observe changes in functional connectivity[6] and brain activation[7]. In addition, non-invasive diffusion MRI has been used to quantify changes in white matter structures[8] by using mathematical modeling to infer both micro and macroscopical changes in tissue based on probing the free random movement of water molecules[9,10]. Abnormalities in diffusion MRI, such as reduced fractional anisotropy (FA), have been shown to be a marker of diffuse demyelination[11] and to be associated with cognitive impairment in patients with MS[12]. As MS is considered to primarily be a brain white matter disease, the focus of diffusion MRI research studies has predominantly targeted structural changes within the white matter[13,14].

The free water fraction (FWF) is another diffusion-based method that aims to disentangle the free water contribution from the diffusion signal[15]. It has previously been used to estimate the fraction of extracellular water in applications such as quantifying the contribution of edema in tumors and the tissue surrounding lesions in MS[16–20]. The FWF can also be estimated from other diffusion models[21]. Fluid accumulation, akin to edema is one of the hallmarks of inflammation and is generally regarded as a first-order immune response[22]. It has been suggested that inflammation and depression may be linked[23], thus, a imaging metric to estimate free water may provide a surrogate biomarker to assess inflammation in vivo.

Depression, anxiety, and fatigue are common among patients with MS, with symptoms that significantly impair patients' quality of life[24–28]. This occurs despite progress in treatments currently available for the disease.

Although living with a disabling disease with increasing loss in motor function, among other life-limiting symptoms, definitely can lead to depression, a more direct impact of subcortical microglial activation, lesion burden, and regional atrophy has also been suggested[24,26,27].

A recent study found that MS patients are more likely to develop clinical depression during pregnancy[29]. Volumetric abnormalities in the basal ganglia has for example recently been demonstrated to be a predictor of fatigue in a large cohort of MS patients[30]. Moreover, the basal ganglia has also been suggested to play a role in depression[31] in general. In studies including diffusion MRI, abnormalities in subcortical gray matter have also previously been observed in MS[32–36]. The subcortical structures including the basal ganglia are therefore particular targets of investigations in this current study.

Similarly, cognitive function is also shown to be affected in patients with MS, at any stage and subtype of the disease[37–41]. The prevalence and severity of cognitive impairment appears greatest in PPMS and SPMS[42]. It has recently been shown in studies including the Brief International Cognitive Assessment for MS (BICAMS) tests[43], that the results on psychometric tests of processing speed as well as verbal and visual memory were correlated with whole brain and gray matter volume measures derived from an MRI examination at baseline[44]. After 2 years, the authors found significant changes in these global volumes that allowed differentiation of patients that were defined as either cognitively impaired or preserved.

Generally, the prevalence of cognitive impairment in MS ranges from 34 to 65% and depression is a symptom in one of four patients between the ages of 18–45[26,45]. Zabad et al. found that patients suffering from PPMS were less at risk to suffer from major depression than RRMS patients[46].

From the studies referred to above, we expect to find associations between MRI-derived measures of subcortical structures and results on cognitive tests and scales assessing fatigue and depression in patients with relapsing-remitting MS. In this study, brain measures are derived from different MRI modalities. Considering the strict interplay described between depression and parenchymal structural changes, a focus will be given to the question whether pathological changes detected by diffusion MRI in the subcortical structures can predict worsening of depression and anxiety symptoms in this patient group.

We find that nearly all subcortical diffusion microstructure estimates at the baseline visit correlate with the depression score at the 2 years follow-up. The predictive nature of baseline free water estimates and depression subscores after 2 years are confirmed in the predictive structural equation modeling analysis with the thalamus showing the greatest effect size. We also find in the general linear model analysis patterns of MRI-free water differences in the thalamus and amygdala/hippocampus area between participants with high and low depression score.

Our findings suggest a relationship between higher levels of free-water in the subcortical structures in an early stage of Multiple Sclerosis and depression symptoms at a later stage of the disease.

## Methods

**Participants**. A total of 65 participants with relapsing-remitting MS as defined by the 2017 revision of the McDonald criteria[2] were recruited at the Department of Neurology, Haukeland University Hospital, based on written informed consent, approved by the Regional Ethics Committee of Western Norway (registration number 2016/31/REK Vest).

$T_1$-weighted ($T_1w$), $T_2$-weighted ($T_2w$), and $T_2$-FLAIR MR imaging were part of a larger imaging protocol acquired in all participants, alongside with a neuropsychological examination of cognitive and emotional function at baseline as well as at two follow-up visits that were scheduled at approximately 1 and 2 years after the baseline examination, respectively. Diffusion-weighted imaging (DWI) was performed as part of the MRI protocol at baseline and at the first follow-up, but not at the second follow-up.

**Clinical examination and testing**. For each visit, a separate clinical examination was scheduled close to the date of the MRI. The clinical examinations included the BICAMS examination comprising the oral part of the symbol digit modalities test (SDMT)[47], the learning trials of both the second edition of the California Verbal Learning Test (CVLT-II)[48] and the Revised Brief Visuospatial Memory Test (BVMT-r)[49]. The BICAMS was developed to provide a short screening procedure for patients with MS[50,51], and has been validated in a Norwegian study[43].

EDSS was used to examine the disability status[3] together with the Fatigue Scale for Motor and Cognitive Functions (FSMC)[52] as well as the Hospital Anxiety and Depression Scale (HADS)[53]. The HADS is a 14-item questionnaire designed to assess the current state of the symptoms, with seven items each for assessment of symptoms related to depression and anxiety. The participant can respond to each question on a 4-point scale after which the points are summed up to give a score between 0 and 21 for each subscale. A total score is calculated as the sum of these two subscales[53]. The scores on the two subscales and the total score are included in the present study.

**MR-protocol**. All MR imaging was performed on a 3T Siemens Prisma system (Siemens Healthineers, Germany) and comprised $T_1w$, $T_2w$, $T_2$-FLAIR, and DWI. The parameters were as follows:

3D volumetric $T_1w$ sagittal volume, TE/TR/TI = 2.28 ms/1.8 s/900 ms, acquisition matrix = $256 \times 256 \times 192$, FOV = $256 \times 256$ mm², slice thickness = 1 mm, 200 Hz/px readout bandwidth and total acquisition duration of 7.4 min.

2D axial $T_2w$ volume, TE/TR = 100.0 ms/6.0 s, acquisition matrix = $512 \times 384$, FOV = $220 \times 220$ mm², slice thickness = 4 mm, 220 Hz/px readout bandwidth, and total acquisition duration of 2.1 min.

3D volumetric $T_2$-FLAIR sagittal volume, TE/TR/TI = 386 ms/5.0 s/1600 ms, acquisition matrix = $256 \times 256 \times 192$, FOV = $256 \times 256$ mm², slice thickness = 1 mm, 751 Hz/px readout bandwidth and total acquisition duration of 6.2 min.

2D axial DWI with 6 diffusion-unweighted volumes and 4 different diffusion-weighted volumes with b-values of 200 (3 directions), 500 (6 directions), 1000 (30 directions), and 2500 (30 directions) s/mm², TE/TR = 82 ms/9 s, acquisition matrix = $128 \times 128$, FOV = $256 \times 256$ mm², slice thickness = 2 mm, 72 slices, 1500 Hz/px readout bandwidth and total acquisition duration of 11.4 min.

**Image processing**. $T_2$ FLAIR images were co-registered to the $T_1w$ images within the same imaging session using SPM12 (UCL, UK). Diffusion-weighted images were motion corrected, masked and eddy current corrected using FSL 6.0.1 (the University of Oxford, UK). FWF maps were created in native space using an in-house routine. In line with previous publications, only b-values < 2000 s/mm² were used[18,19,54]. The resultant parametric maps from the diffusion imaging were subsequently also co-registered to the corresponding $T_1w$ image.

MS brain lesions were outlined by icobrain ms (Icometrix, Belgium), an FDA-approved and CE-marked radiological services provider. Subcortical structures were segmented on the $T_1w$ images using the FSL FIRST tool after lesion filling had been performed. The segmented structures included separate measures for both left and right hemispheres of the following subcortical structures: thalamus, caudate, putamen, pallidum, amygdala, and accumbens. Measures over both hemispheres included whole white and gray matter, combined brain stem and fourth ventricle area and hippocampus. For analysis, the measures for left and right hemispheres of the subcortical structures were averaged. In the whole white and gray matter FWF analysis, areas with lesions were excluded by artificially extending each lesion with a $5 \times 5 \times 5$ mm³ Gauss filter and subtracting it from the analysis mask. For the whole white and gray matter volume analysis, lesions were filled-in before measures were computed.

As well as computing total volume and volume changes over time, segmented structures were used for masking regions of interest (ROI) in the free water maps.

To account for partial volume effects in these ROI masks when estimating free water, their size was reduced by applying the erosion function imerode in Matlab 9.5.0 (the MathWorks, Natick, MA). The strength of the erosion was set to be approximately 1/3 of total volume before erosion.

For the general linear model (GLM) analysis, individual $T_1w$ images were transformed into standard space based on the MNI152 $T_1w$ template. The transformation was then applied to the FWF maps from the first visit and images were subsequently smoothed with a $3 \times 3 \times 3$ mm³ Gaussian filter.

**Statistical analysis**. Statistical analysis was performed in Matlab 9.5.0 (the MathWorks, Natick, MA). Pearson correlations were computed between MRI data (total white and gray matter, subcortical structures volume, volume change between timepoints, FWF) and all test scores and responses on the questionnaires. Total number of test variables in the correlation matrix was 18 with 46 observations each. Resultant p-value matrices were corrected for multiple comparisons using false discovery rate (FDR) testing and only FDR-corrected p-values are reported[55,56]. A correlation coefficient greater than $r = \pm 0.3$ with a $P$-value < 0.05 (FDR-corrected) was considered significant.

To investigate whether FWF at the first visit can predict HADS and the other test and questionnaire scores at the most recent clinical assessment (the third visit), structural equation modeling (SEM) was set up using a partial least squares algorithm (PLS-SEM)[57]. For this, the 18 measurement variables belonging to the FWF mean values of each subcortical region and individual test and questionnaire scores were grouped into the following five latent constructs: FWF in subcortical region (including thalamus, caudate, putamen, pallidum, amygdala, and accumbens), EDSS, BICAMS (including SDMT, CVLT-II and BVMT-r), FSMC (including cognitive, motor and total fatigue scores) and HADS (including both the anxiety, depression and a total anxiety/depression score). The model was set up as a formative measurement model.

For the GLM analysis focusing on symptoms of depression at the last clinical visit, subjects were grouped by their HADS depression subscore 0–2 ($n = 23$) and 2–9 ($n = 23$) as recorded at the 2-year follow-up and a two-sample t-test was performed to the FWF maps of the baseline visit in SPM12 (UCL, London, UK) with age as a covariate. The cut-off value of HADS depression subscore was chosen as it provided equal sample sizes.

**Reporting summary**. Further information on research design is available in the Nature Portfolio Reporting Summary linked to this article.

## Results

Out of the 65 recruited subjects with relapsing-remitting MS, two opted out and withdrew their consent, two did not attend the baseline visit, four did not return after the baseline visit and seven did not attend for the 2-year follow-up. One subject missed the 1-year follow-up but had completed the baseline and 2-year follow-up visits, and was not removed from the dataset. A further three subjects had incomplete cognitive scores at the 2-year follow-up visit and were therefore excluded. In total, 46 participants had complete data collection at baseline and 2-year follow-up and

**Table 1 Study participant demographics.**

| Characteristic | Baseline | 1-year follow-up | 2-year follow-up | 1-year follow-up HADS D < 2 | 2-year follow-up HADS D > 2 |
|---|---|---|---|---|---|
| Number of participants | 46 | 45 | 46 | 23 | 23 |
| Age (years) | | | | | |
| Median | 37.9 | 38.8 | 39.9 | 36.9 | 41.9 |
| Range | 21.0–63.8 | 21.9–64.8 | 22.8–65.9 | 26.5–55.9 | 22.9–65.9 |
| Sex | | | | | |
| Female | 32 | 31 | 32 | 17 | 15 |
| Male | 14 | 14 | 14 | 6 | 8 |
| EDSS | 1.3 ± 0.9 | 1.5 ± 0.8 | 1.5 ± 0.9 | 1.3 ± 0.8 | 1.8 ± 0.9 |
| FSMC | | | | | |
| Motor | 14.1 ± 9.7 | 14.8 ± 10.4 | 15.5 ± 10.7 | 9.9 ± 8.9 | 21.0 ± 9.7 |
| Cognitive | 14.2 ± 10.5 | 14.8 ± 10.8 | 14.9 ± 9.9 | 9.4 ± 8.5 | 20.4 ± 8.0 |
| Total | 28.3 ± 19.6 | 29.6 ± 20.9 | 30.4 ± 20.1 | 19.3 ± 16.8 | 41.5 ± 17.0 |
| HADS | | | | | |
| Anxiety | 5.0 ± 3.8 | 4.3 ± 2.7 | 4.6 ± 3.2 | 2.9 ± 2.7 | 6.3 ± 3.0 |
| Depression | 2.7 ± 3.0 | 2.2 ± 2.2 | 2.7 ± 2.6 | 0.6 ± 0.6 | 4.7 ± 2.3 |
| Total | 7.7 ± 6.0 | 6.4 ± 4.1 | 7.2 ± 5.2 | 3.5 ± 3.0 | 11.0 ± 4.3 |
| BICAMS | | | | | |
| SDMT | 56.1 ± 10.7 | 58.5 ± 12.0 | 58.8 ± 11.7 | 61.9 ± 9.8 | 55.8 ± 12.9 |
| CVLT-II | 55.4 ± 10.9 | 57.8 ± 8.7 | 58.2 ± 10.4 | 61.8 ± 9.5 | 54.6 ± 10.1 |
| BVMT-r | 27.4 ± 5.3 | 28.3 ± 5.95 | 27.3 ± 5.6 | 28.1 ± 4.7 | 26.4 ± 6.3 |

*EDSS* Expanded disability status scale. *FSMC* Fatigue scale for motor and cognitive functions, *HADS* Hospital anxiety and depression scale, *BICAMS* Brief international cognitive assessment for Multiple Sclerosis, *SDMT* Symbol digit modalities test, *CVLT-II* the learning trials of the 2nd edition of the California Verbal Learning Test, *BVMT-r* Revised brief visuospatial memory test.

were therefore included in this study, see Table 1 for participant demographics. MRI alongside with a neuropsychological examination of cognitive and emotional function at baseline as well as at two follow-up visits were scheduled at approximately 1 and 2 years after the baseline examination, respectively. The average time between MRI and neurocognitive assessment was 7.5 ± 31.0 days.

In the following, if not stated otherwise, all results are reported after multiple comparison testing corrections.

Mean values of FWF for each visit and ROI before and after ROI erosion are given in Supplementary Table 1.

**Association of FWF in subcortical structures and depression and anxiety score (HADS).** FWF at baseline did not correlate with the scores on any of the two HADS subscales at baseline or the 1-year follow-up, however FWF at baseline was correlated with the combined (anxiety and depression) HADS scores at the 2-year follow-up in the: Thalamus ($r = 0.40$, $P = 0.01$), putamen ($r = 0.32$, $P = 0.03$), pallidum ($r = 0.37$, $P = 0.01$), hippocampus ($r = 0.43$, $P = 0.004$), amygdala ($r = 0.41$, $P = 0.01$), and accumbens ($r = 0.40$, $P = 0.01$).

**Association of FWF in subcortical structures and depression subscore.** FWF at baseline was also correlated with the HADS depression subscore at the 2-year follow-up in the: Thalamus ($r = 0.47$, $P = 0.002$), putamen ($r = 0.40$, $P = 0.01$), pallidum ($r = 0.42$, $P = 0.005$), hippocampus ($r = 0.42$, $P = 0.005$), amygdala ($r = 0.34$, $P = 0.02$), and accumbens ($r = 0.44$, $P = 0.003$).

**Testing for a relationship between depression and FWF in subcortical structures using PLS-SEM and GLM analysis.** The PLS-SEM model converged in 301 iterations and fulfilled construct reliability of all latent variables with a Cronbach Alpha >0.7. Predictive nature of baseline diffusion measures and the HADS depression subscore was confirmed in the SEM analysis (total effect size of path: 0.46).

Figure 1 shows the results of the two-sample t-test GLM analysis. Clusters of high t-values on this map highlight

significantly different ($P = 0.001$, uncorrected) areas in the FWF maps between participants with low (<2, $n = 23$) vs. higher (>2, $n = 23$) HADS depression subscore. Clusters are clearly discernable in the amygdala, hippocampus, and thalamus. Additional clusters were also seen in the corpus callosum, cerebellar vermis, and precuneus.

**Association of FWF in subcortical structures and anxiety subscore.** FWF at baseline was correlated with the HADS anxiety subscore at the 2-year follow-up between the: Hippocampus ($r = 0.35$, $P = 0.02$) and amygdala ($r = 0.38$, $P = 0.01$).

In summary, FWF in all the segmented subcortical structures except for the caudate were correlated with one of the HADS metrics. FWF in the subcortical structures was also not correlated to any other clinical measurement at any timepoint.

**Association of FWF in total white and gray matter and cognitive measures (BICAMS).** Diffusion microstructure in cortical gray matter was found to be negatively correlated with all BICAMS metrics at their respective timepoints: r-BVMT (baseline visit: $r = -0.46$, $P < 0.001$, 1-year follow-up: $r = -0.41$, $P < 0.001$), SDMT (baseline visit $r = -0.58$, $P < 0.001$, 1-year follow-up visit: $r = -0.54$, $P < 0.001$) the California Verbal Learning Test-II (CVLT-II – baseline visit: $r = -0.37$, $P = 0.001$, 1-year follow-up visit: $r = -0.33$, $P = 0.004$). Diffusion microstructure in the subcortical structures and whole gray matter was not found to be correlated with any of the BICAMS test metrics.

**Association of volumetric measures and cognition (BICAMS) and fatigue (FSMC).** Hippocampal and accumbens volumes at baseline were positively correlated with individual BICAMS metrics such as the SDMT ($r = 0.49$, $P < 0.01$) and the r-BVMT ($r = 0.35$, $P = 0.02$). This relationship was maintained in the 1-year follow-up visit were the hippocampus volume again correlated with SDMT ($r = 0.37$, $P = 0.03$) and the accumbens volume with r-BVMT ($r = 0.36$, $P = 0.04$). At the 2-year follow-up, SDMT remained moderately correlated with hippocampus

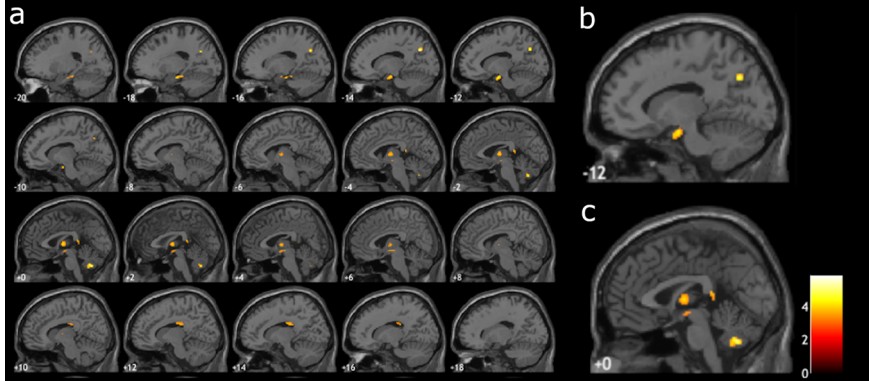

**Fig. 1 Resultant t-map ($P = 0.001$, uncorrected) of the GLM analysis of FWF maps at the baseline visit between subjects with HADS depression <2 ($n = 23$) and HADS depression >2 ($n = 23$) at the 2-year follow-up visit overlaid on the sagittal MNI152 T1-weighted template. a** Clusters highlight differences between the groups and are seen in the amygdala and hippocampus area (slices −20 to −10). Clusters are also seen in the thalamus area (slices −6 to +6) Additional clusters were seen in the corpus callosum (slices −10 to +16), cerebellar vermis area (slices −4 to +2) and precuneus (slices −20 to −10). **b** Magnification of slice −12 highlighting the clusters in the amygdala and hippocampus area. **c** Magnification of slice 0 highlighting the cluster in the thalamus area.

volume ($r = 0.40$, $P = 0.02$) and accumbens volume ($r = 0.48$, $P = 0.003$).

At the 2-year follow-up, thalamus volumes were correlated with cognitive fatigue ($r = -0.37$, $P = 0.03$) and combined fatigues score for motor and cognitive functions ($r = -0.35$, $P = 0.04$). The hippocampus ($r = -0.43$, $P = 0.01$) as well as the amygdala ($r = -0.37$, $P = 0.03$) were correlated with cognitive fatigue subscores. The hippocampus volume was also correlated with the motor subscore from the fatigue scale ($r = -0.35$, $P = 0.04$) and the total fatigue scores ($r = -0.40$, $P = 0.02$).

Results from the volumetric analysis of the subcortical structures can be found in Supplementary Table 2 with atrophy measures in percent in Supplementary Tables 3 and 4. No statistically significant correlations between atrophy as a percentage-change and neuropsychological test scores were found.

## Discussion

Changes in cognitive and motor function in MS are most generally recorded using a palette of different tests that aim to record different sets of performances. Such tests include the EDSS scale[3], 9-hole peg test[58], timed 25-foot walk, and/or more specialized tests focusing on the auditory and visual system[59,60]. In addition, quality of life, mood, and mental health[61,62] may be recorded with additional questionnaires that aim to register symptoms such as fatigue, pain, and depression.

Recommendations have been made on which assessment tools to include in monitoring of MS in standard practice and in clinical trials. The administration of these tools can, however, be time-consuming. Key targets of shortened exams, such as the BICAMS test[50,63], are to cover a broad-enough range of cognitive domains to be sensitive enough to monitor cognitive decline and changes in relation to treatment. We believe there is benefit in adding questionnaires such as the HADS and FSMC. The HADS has been widely-used and by using well-defined score cut-offs for depression (above 8 out of a possible 21 a.u.), a meta-analysis of 747 publications found a sensitivity and specificity of HADS of approximately 0.80 according to the Cronbach alpha[64].

We observed significant correlations between free-water in the subcortical structures and the HADS depression and anxiety scores. As a means to correct for partial volume, we used erosion to shrink the ROIs, thereby minimizing the effect of neighboring structures and tissues to our results. The FWF diffusion metric for a given ROI increased after erosion, implying that our findings are not related to increased diffusion due to partial volume with cerebrospinal fluid in the extracellular space that may have increased due to atrophy. If partial volume and atrophy were indeed a driver, we would have expected decreased FWF values after erosion. We therefore suggest that our observation is instead due to changes within the subcortical structures that could be driven by gray matter changes in those areas. It would be beneficial to investigate for the presence of subcortical gray matter lesions; however, identification of cortical and subcortical gray matter lesions in standard MR images is difficult[65]. It has been shown that cortical gray matter aberrations are present in the earliest stages of MS[66] and a post mortem study did confirm that both focal demyelinating lesions and diffuse neurodegeneration are common in the deep gray matter of MS patients[67]. The latter study also found that demyelination was most present in the hippocampus and caudate and could already be seen at an early disease stage[67]. An investigation into demyelination and lesions in the deep gray matter would be of future interest using specialized MRI sequences such as double inversion recovery imaging (DIR) that can aid gray matter lesion detection[65].

We observed correlations between our free water diffusion metric and both the anxiety and depression subscores in nearly all subcortical structures. Hyperactivity of the hypothalamic-pituitary-adrenal axis has been suggested to be an endocrine basis for the development of depression[26,68] and it is therefore not surprising to see the hippocampus and thalamus presenting with increased free-water, as seen in our study.

Only the caudate did not exhibit any correlation between the depression subscore and free water diffusion index. This is consistent with earlier research, suggesting that the caudate does not play a role in mood, depression or fatigue, but is rather associated with motor function and cognitive processes such as memory and learning[69–71].

In our sample, HADS depression scores at baseline were $2.7 \pm 3.0$ out of possible 21 points and $5.0 \pm 3.8$ for anxiety, again out of possible 21 points and on the 2-year follow-up $2.7 \pm 2.6$ and $2.9 \pm 2.7$ for depression and anxiety, respectively. However, the theoretical upper maximum of 21 points is rarely scored, a cut-off of >7 is usually used to define a subject as suffering clinically from depression or anxiety[64]. Our sample therefore represents a good range between mild symptoms and what would qualify as clinical depression. Our results suggest that FWF may be a sensitive enough technique to detect depression early.

We employed false discovery rate testing as a means to correct for multiple comparisons. We chose this method as it has been

previously employed[30] in this type of study and since Bonferroni-Holm or permutations tests were deemed unsuitable for this dataset: While the Bonferroni-Holm method penalizes a large number of comparison variables, permutation testing is more suitable for larger datasets such as those from genome sequencing[72]. To test whether baseline visit FWF could be used to predict HADS, we used a PLS-SEM analysis which overall confirmed the link between FWF and HADS. To move away from a ROI-based analysis we also employed whole-brain GLM voxel-wise statistical testing that also highlighted statistically significant differences in only a few distinct regions with strong clusters in the amygdala, hippocampus, and thalamus, similar to our ROI-based correlation analysis results. This also highlights that not only are there differences in the subcortical structures that predict depression symptoms, but also that these are varied and their strength can clearly differentiate the cohort into two groups. The GLM analysis results did unfortunately not survive FDR correction, possibly due to the reduction in sample size by splitting our cohort into two sub-groups. Aside from the clusters seen in the amygdala, hippocampus, and thalamus, we also observed clusters in the corpus callosum, precuneus, and cerebellum. While we attribute the cluster in the corpus callosum and precuneus possibly to artefacts, the strength and distinctiveness of the cluster in the cerebellum is striking and requires further investigation.

All of the individual metrics of the BICAMS test battery correlated with the free water diffusion index in the cortical gray matter at the respective timepoints. A recent study by Genç et al.[73] that employed a diffusion-based neurite density metric in 498 participants demonstrated a strong association between cortical gray matter and performance on an IQ test. This may imply that diffusion in the cortical gray matter may also influence the performance in the tasks included in the BICAMS test battery.

In addition to the diffusion findings, we found a significant correlation between the results on the SDMT and hippocampus volumes at each of the respective timepoints and the r-BVMT to be correlated with the accumbens volume at both baseline and 1-year follow-up. The role of the hippocampus has been linked to memory, inhibition, and spatial cognition and has been related to cognitive changes in diseases like Huntington's and Alzheimer's disease[74]. Our findings may be explained by characteristics of the SDMT: In the SDMT task, the examinee is asked to match numbers with geometrical shapes during a period of 90 s, and by this putting a strong load on both memory function and spatial cognition. The r-BVMT is characterized by memory and learning components[49,59]. The correlation between r-BVMT and accumbens may therefore be attributed to the accumbens' contribution to learning, memory, and reward mechanisms, as was recently demonstrated in a model of dopaminergic neuron loss[75].

A recent cross-sectional study by Fleischer et al.[30] demonstrated that subcortical volumes could be used as an early predictor of fatigue in MS. Fleischer et al.[30] found mainly that the caudate volume at baseline correlated with fatigue at a follow-up visit 4 years later. In our study, we saw correlations at the 2-year follow-up between cognitive fatigue and the thalamus, hippocampus, and amygdala. While we did not see a correlation between the caudate volume and fatigue, the highest correlation we observed was in the hippocampus ($r = -0.43$, $P = 0.01$), which the author of the Fleischer et al.[30] study suggested is involved mainly at the onset of MS. We therefore attribute the lack of caudate findings to the short disease duration in our cohort.

FWF is a diffusion microstructure parameter that can be computed in a few seconds from standard diffusion tensor imaging (DTI) acquisitions. Many centers include DTI acquisitions to facilitate fractional anisotropy (FA) and DTI tractography

estimation, making all these studies potentially compatible with FWF estimation. More complex models that require longer, dedicated acquisitions such as restricted spectrum imaging (RSI)[76] or neurite orientation dispersion and density imaging (NODDI)[77] also include an isotropic diffusion estimate which should provide similar information to FWF. It may therefore be possible to reproduce our results with one of these models.

Our data may suggest that processes related to MS in the subcortical structures may contribute to the development of depression symptoms. If early changes in FWF can predict onset and severity of depression, clinical decisions can be taken to prepare the patients and their families with complimentary treatment for depression such as cognitive behavioral therapy.

## Data availability

The data that support the findings of this study are available on request from the corresponding author. The data are not publicly available due to containing information that could compromise the privacy of research participants. Approval for data sharing is subject to approval by the author's local ethics committee and a formal data sharing agreement.

## Code availability

The code used in the analysis has been made available online as Supplementary Data.

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

## Acknowledgements

The authors would like to thank the MS patients for participation in the study, and express their thanks to the Health Authorities of Western Norway for funding of the project. We are grateful to Nuno Pedrosa de Barros from Icometrix for providing the lesion analyses and to scientists Hauke Bartsch and Erling Andersen, Haukeland University Hospital, for their involvement in MRI data collection, handling and storage. We are thankful to the team of neuroradiologists at the Department of Radiology, Haukeland University Hospital, for MR image reading and reporting. We are obliged to MS nurses Anne-Britt Rundhovde Skår and Randi C. Haugstad at the Norwegian MS Competence Center, Haukeland University Hospital, for their invaluable contribution to the study, and to health secretary Angunn Solberg at the outpatient clinic at the Department of Neurology, Haukeland University Hospital, for her contribution to logistics. MMIV is jointly hosted by Haukeland University Hospital and University of Bergen and supported as a centre by grants from the Trond Mohn Foundation. Neuro-SysMed is jointly hosted by Haukeland University Hospital and University of Bergen and supported as a centre for Clinical Treatment Research (FKB) by grants from The Research Council of Norway, project number 288164. Ellen Skorve has received majority of funding through PhD-scholarship from the Health Authorities of Western Norway (3-year fellowship). Additional financial support for the MRI investigation study was provided by Dr. Niels Vilhelm Henrichsen and wife Anna Henrichsens Legacy Fund.

## Author contributions

All authors contributed to the conception of the study and the study design. F.R. carried out the analysis and wrote the manuscript. E.S. and A.L. collected the data and contributed to analysis. O.P. developed the free water model and contributed to data analysis. F.Z. suggested the prediction modeling. A.L., Ø.T., K.M.M., and R.G. provided supervision and contributed to edits of the manuscript. All authors provided feedback on the manuscript.

## Funding

## Competing interests

The authors declare the following competing interests: Ellen Skorve has received initial funding for this study through an unrestricted research grant from Novartis (project planning and inclusion phase). Øivind Torkildsen has received speaker honoraria from and served on scientific advisory boards for Biogen, Sanofi-Aventis, Merck, and Novartis. Kjell-Morten Myhr has received unrestricted research grants to his institution, scientific advisory board or speaker honoraria from Biogen, Merck, Novartis, Roche, and Sanofi; and has participated in clinical trials organized by Biogen, Merck, Novartis, Roche, and Sanofi. All other authors report no competing interests.
