## [Peer Review File · Communications Medicine]

Reviewers' comments:

Reviewer #1 (Remarks to the Author):

In this manuscript the authors assess links between early microstructural changes in subcortical volumes & depressive symptoms in a longitudinal study of 46 people with relapsing-remitting MS. Free water fraction was used to assess tissue microstructure, and a binary split was used to classify patients as having 'high' or 'low' depression scores. They find that all subcortical diffusion measures, except caudate, correlated with depression at 2 year visit. Correlations were carried out with depression, anxiety and a combined depression+anxiety score. The authors conclude that higher levels of free water in subcortical structures is linked to depression early in the course of MS (but not cognitive processes).

The study is interesting and the methods have been well carried out. Recruiting and retaining participants for these longitudinal studies is challenging – so the authors should be commended. There are some issues about the choice of outcome variables and whether the introduction perhaps misses out other studies of free water diffusion in MS. But otherwise the study reads well and is useful.

1. In the introduction the authors state that free water fraction has previously been used in cancer imaging, but not in MS. However axonal water fraction can be derived from diffusion kurtosis imaging models – and has been used a few times in people with MS (see Hori et al., 2022 Mag Res in Med Sci). It might be worth mentioning these other approaches here.
2. Also in the intro, the authors start mentioning links between the BICAMS and grey matter volumes in MS – but this trails off without noting what correlations were found. Can they present more details here?
3. Brain lesions were automatically segmented but it's not clear whether lesion-filling was carried out before estimating volumes?
4. Do the authors really believe it is useful to combine Anxiety and Depression scores? Given that both have separate anatomical bases it's not clear what's gained by using this summed measure as an outcome.

5. The authors test for association between free water diffusion in the whole grey matter and cognitive performance – but not with free water diffusion in individual grey matter structures. Was this because no associations were found? If so, it would be helpful to report this and consider in the discussion.

Reviewer #2 (Remarks to the Author):

What are the major claims of the paper? Are they novel and will they be of interest to others in the community and the wider field?

1) This paper will definitely be vital to MS research and the wider community. In a nutshell, the paper seeks to characterise neuroimaging correlates of affective and cognitive presentations in MS. Considering the high prevalence of some of these disorders in MS and the poorer outcomes and prognosis from existing co-morbidity, such research is vital not only for better understanding of etiopathology (of both MS and affective disorders) but also could be important in the identification of therapeutic end points during large scale clinical trials. Indeed, when MS occurs along with some of these presentations it is difficult to treat.

Is the work convincing, and if not, what further evidence would be required to strengthen the conclusions? On a more subjective note, do you feel that the paper will influence thinking in the field? Please feel free to raise any further questions and concerns about the paper.

2) In addition to free water fraction (FWF), perhaps also considering other diffusion parameters such as MD, FA, RD, AD, as well as NODDI parameters (ODI, ICFV, and ISOVF) may have made for stronger conclusions but that would also mean modifying methodology.

We would also be grateful if you could comment on the appropriateness and validity of any statistical analysis, as well as the ability of a researcher to reproduce the work, given the level of detail provided.

3) Statistical applications appear okay, and can be reproducible.

4) Further reviews attached (in Word document)

Microstructural changes precede depression in patients with relapsing-remitting Multiple Sclerosis

Frank Riemer^{1,2}, Ellen Skorve^{2,3}, Ofer Pasternak^{4,5}, Fulvio Zaccagna^{6,7}, Astri J Lundervold⁸, Øivind Torkildsen^{2,3}, Kjell-Morten Myhr^{2,3} and Renate Grüner^{1,9}

Abstract

Multiple Sclerosis lesions in the brain and spinal cord **can** lead to different symptoms, including cognitive and mood changes. In this study we explore the temporal relationship between early **microstructural changes in subcortical volumes** and cognitive and emotional function in a longitudinal cohort study of patients with relapsing-remitting Multiple Sclerosis (**RRMS**). A specific focus will be given to **brain imaging predictors of future depression**.

In vivo imaging in **forty-six patients with RRMS** was performed annually over three years using an extensive magnetic resonance imaging protocol. In parallel, patients were assessed with the Brief International Cognitive Assessment for Multiple sclerosis, the Expanded Disability Status Scale, the **Hospital Anxiety and Depression Scale** and the **Fatigue Scale for Motor and Cognitive Functions**. We were particularly interested to explore if there was a relationship between imaging markers at baseline and onset of depressive symptoms at a later stage of the disease. From these data, microstructural changes were estimated in subcortical structures using the **free water fraction**, a diffusion-based MRI metric. These brain imaging measures were then correlated with results on the cognitive assessments scores. Statistical analyses were corrected for multiple comparisons using false discovery rate testing and a bias for selection of region-of-interest and age were employed as a co-variate to account for the effects of partial volume and atrophy due to normal aging. Predictive structural equation modeling was set up to further explore the relationship between imaging and the assessment scores. In a general linear model analysis, the cohort was split into patients with higher and lower depression scores.

All subcortical diffusion microstructure estimates at the baseline visit correlated with the depression score at the two years follow-up except for the caudate: In particular, moderate

correlations were found between depression subscore and free water in the thalamus, the putamen, hippocampus and accumbens. In addition, the combined anxiety and depression score were correlated with free water fraction measures of the hippocampus, amygdala and accumbens.

The predictive nature of baseline free water estimates and depression subscores after two years were tested for and confirmed in the predictive structural equation modeling analysis with the thalamus again showing the greatest effect size. The general linear model analysis showed patterns of MRI free water differences in the thalamus and amygdala/hippocampus area between participants with high and low depression score.

Our data suggests a relationship between higher levels of free-water in the subcortical structures in an early stage of Multiple Sclerosis and depression symptoms at a later stage of the disease. We hypothesize a physiological or pathological process in the basal ganglia and related structures to significantly contribute to mood disorders in Multiple Sclerosis.

Author affiliations:

1 Mohn Medical Imaging and Visualization Centre (MMIV), Department of Radiology, Haukeland University Hospital, 5021 Bergen, Norway

2 Neuro-SysMed, Department of Neurology, Haukeland University Hospital, 5053 Bergen, Norway

3 Department of Clinical Medicine, University of Bergen, 5020 Bergen, Norway

4 Department of Psychiatry, Brigham and Women's Hospital, Harvard Medical School, Boston, MA, 02215

5 Department of Radiology, Brigham and Women's Hospital, Harvard Medical School, Boston, MA, 02215

6 Department of Biomedical and Neuromotor Sciences, Alma Mater Studiorum, University of Bologna, 40125 Bologna, Italy

7 IRCCS Istituto delle Scienze Neurologiche di Bologna, Functional and Molecular Neuroimaging Unit, 40139 Bologna, Italy

8 Department of Biological and Medical Psychology, University of Bergen, 5009 Bergen, Norway

9 Department of Physics and Technology, University of Bergen, 5007 Bergen, Norway

Correspondence to: Frank Riemer

Medical Imaging and Visualization Centre

Haukeland University Hospital

Dept of Radiology

Post Office Box 1400

N – 5021 Bergen

f.riemer@web.de

Running title: Free water MS diffusion paper

Keywords: Multiple sclerosis; MRI; Depression; Inflammation; Subcortical

Abbreviations: 2D = Two-dimensional; 3D = three-dimensional; BICAMS = Brief International Cognitive Assessment for Multiple Sclerosis; BVMT-r = Revised brief visuospatial memory test; CE = Conformite Europeenne; CSF = Cerebrospinal fluid; CVLT-II = Second edition of the California Verbal Learning Test; DIR = Double inversion recovery;

DTI = Diffusion tensor imaging; DWI = Diffusion-weighted imaging; EDSS = Expanded Disability Status Scale; FA = Fractional anisotropy; FDA = Food and Drug Administration; FDR = False discovery rate; FOV = Field-of-view; FSMC = the Fatigue Scale for Motor and cognitive Functions; FWF = Free water fraction; GLM = General linear model; HADS = Hospital Anxiety and Depression Scale; IQ = Intelligenzquotient; MNI152 = Montreal National Institute 152-subject template; NODDI = Neurite orientation dispersion and density imaging; PLS-SEM = Partial least squares structural equation modeling; px = pixel; ROI = Region-of-interest; RSI = Restricted spectrum imaging; SDMT = Symbol digit modalities test; SEM = Structural equation modeling; T_{1w} = T₁-weighted; T₂-FLAIR = T₂-weighted fluid-attenuated inversion recovery; T_{2w} = T₂-weighted; TE = Echo time; TI = Inversion time; TR = Recovery time.

Introduction

Multiple Sclerosis (MS) is an inflammatory and degenerative central nervous system disease characterized by demyelination, axonal loss and gliosis.¹ Clinically, the disease is typically associated with impairment of motor and sensory function, but also cognitive and emotional functions that commonly lead to progressive disability with reduced quality of life and shortened life expectancy. The diagnosis is based on the history of symptoms, clinical findings, and measures derived from MRI examinations and cerebrospinal fluid (CSF) analyses.² Most patients (85-90%) experience a relapsing-remitting disease course, and 10-15% experience a primary progressive course. New disease activity is most often visible as lesions on MRI images, and brain changes are expected to be present before the core clinical symptoms of the disease. In the present longitudinal study, we investigate associations between the brain measures and clinical symptoms in a group of **recently diagnosed patients with MS**.

Longitudinal recording of clinical relapses, disability progression evaluated by the Expanded Disability Status Scale (EDSS),³ as well as the identification of new lesions on structural MRI images² are key elements when identifying and monitoring patients with MS. In addition to identifying lesions, MR research has used basic science approaches to better understand the pathophysiology of the disease, and to provide complimentary measures of inflammation,⁴ the rate of brain atrophy through volumetry,⁵ and changes in functional connectivity⁶ and brain

activation⁷ through Fmri (~~reword sentence~~). In addition, non-invasive diffusion MRI has been used to quantify changes in white matter structures⁸ by using mathematical modeling to infer both micro and macroscopical changes in tissue based on probing the free random movement of water molecules^{9,10}. Abnormalities in diffusion MRI, such as reduced fractional anisotropy (FA), have been shown to be a marker of diffuse demyelination¹¹ and to be associated with cognitive impairment in patients with MS.¹² As MS is considered to primarily be a brain white matter disease, the focus of diffusion MRI research studies has predominantly targeted structural changes within the white matter.^{13,14}

The free water fraction (FWF) is another diffusion-based method that aims to disentangle the free water contribution from the diffusion signal.¹⁵ It has previously been used to estimate the fraction of extracellular water in applications such as quantifying the contribution of edema in tumors,¹⁶⁻¹⁹ but not in MS. Fluid accumulation, akin to edema is one of the hallmarks of inflammation and is generally regarded as a first order immune response.²⁰ It has been suggested that inflammation and depression may be linked,²¹ thus, a imaging metric to estimate free water may provide a surrogate biomarker to assess inflammation *in vivo*.

Depression, anxiety and fatigue are common MS symptoms that significantly impair patients' ~~the~~ quality of life,²²⁻²⁶ despite progress in treatments currently available for the disease (~~reword~~). A recent study found that MS patients are more likely to develop clinical depression during pregnancy.²⁷ While living with a disabling disease with increasing loss in motor function, among other life-limiting symptoms, can lead to depression, it has also been suggested that subcortical microglial activation, lesion burden and regional atrophy play a role (~~reword~~).^{22,24,25} Volumetric abnormalities in the basal ganglia has recently been demonstrated to be a predictor of fatigue in a large cohort of MS patients.²⁸ Moreover, the basal ganglia has also been ~~suggested~~ ~~implied~~ to play a role in depression²⁹ in general. Using diffusion MRI, abnormalities in subcortical gray matter have also previously been observed in MS.³⁰⁻³⁴ Therefore, these structures (~~name the structures you are referring to, and reword~~) are particular targets of investigations in this current study.

~~Similarly, c~~Cognitive function is also shown to be affected ~~already~~ in patients with MS, at any stage and subtype of the disease.³⁵⁻³⁹ (~~please confirm this; may be more prevalent in PPMS~~ ~~ref~~). This has recently been shown in studies including the Brief International Cognitive

Assessment for MS (BICAMS) tests,⁴⁰ and the results on subtests from this test battery were correlated with whole brain and grey matter volumes derived from an MRI examination.⁴¹

From the studies referred to above, we expect to find associations between MRI derived measures of subcortical structures and results on cognitive tests and scales assessing fatigue and depression in patients with relapsing-remitting MS. In this study, brain measures will be derived from different MRI modalities. Considering the strict interplay described between depression and parenchymal structural changes, a focus will be given to the question whether pathological changes detected by diffusion MRI in the subcortical structures could predict worsening of depression and anxiety symptoms in this patient group (may also require some rewording here using linking statements).

Commented [NP(1): Introduce diffusion MRI in earlier paragraphs, include prevalence values for both cognitive impairment and depression in MS and explore literature that includes phenotypic variations. Add some phenotypic /clinical differences between RRMS and PPMS

Commented [NP(2): These paragraphs may require some restructuring. See next comment.

Materials and methods

Participants

A total of 65 participants with relapsing-remitting MS as defined by the 2017 revision of the McDonald criteria² were recruited at the Department of Neurology, Haukeland University Hospital, based on written informed consent, approved by the Regional Ethics Committee of Western Norway (registration number 2016/31/REK Vest).

Commented [NP(3)]: 46 participants were mentioned in Abstract.

T₁-weighted (T_{1w}), T₂-weighted (T_{2w}) and T₂-FLAIR MR imaging were part of a larger imaging protocol acquired in all participants, alongside with a neuropsychological examination of cognitive and emotional function at baseline as well as at two follow-up visits that were scheduled at approximately one and two years after the baseline examination respectively. A diffusion weighted imaging (DWI) examination was performed at baseline and at the first follow-up (one year), but not at the second follow-up (the final timepoint after two years from baseline).

Commented [NP(4)]: Not sure I understand this statement

Clinical examination and testing

For each visit, a separate clinical examination was scheduled close to the date of the MRI. The clinical examinations included the BICAMS examination comprising the oral part of the symbol digit modalities test (SDMT),⁴² the learning trials of both the second edition of the California Verbal Learning Test (CVLT-II)⁴³ and the Revised Brief Visuospatial Memory Test (BVMT-r).⁴⁴ The BICAMS was developed to provide a short screening procedure for patients with MS,^{45,46} and has been validated in a Norwegian study.⁴⁰

EDSS was used to examine the disability status³ together with the Fatigue Scale for Motor and Cognitive Functions (FSMC)⁴⁷ as well as the Hospital Anxiety and Depression Scale (HADS).⁴⁸ The HADS is a 14-item questionnaire designed to assess the current state of the symptoms, with seven items each for assessment of symptoms related to depression and anxiety. The participant can respond to each question on a 4-point scale after which the points are summed up to give a score between 0 and 21 for each subscale.⁴⁸

MR-protocol

All MR imaging was performed on a 3T Siemens Prisma system (Siemens Healthineers, Germany) and comprised T_{1w}, T_{2w}, T₂-FLAIR and DWI. The parameters were as follows:

3D volumetric T_{1w} sagittal volume, TE/TR/TI = 2.28 ms/1.8 s/900 ms, acquisition matrix = 256 x 256 x 192, FOV = 256 x 256 mm², slice thickness = 1 mm, 200 Hz/px readout bandwidth and total acquisition duration of 7.4 minutes.

2D axial T_{2w} volume, TE/TR = 100.0 ms/6.0 s, acquisition matrix = 512 x 384, FOV = 220 x 220 mm², slice thickness = 4 mm, 220 Hz/px readout bandwidth and total acquisition duration of 2.1 minutes.

3D volumetric T₂-FLAIR sagittal volume, TE/TR/TI = 386 ms/5.0 s/1600 ms, acquisition matrix = 256 x 256 x 192, FOV = 256 x 256 mm², slice thickness = 1 mm, 751 Hz/px readout bandwidth and total acquisition duration of 6.2 minutes.

2D axial DWI with 6 diffusion-unweighted volumes and 4 different diffusion-weighted volumes with b-values of 200 (3 directions), 500 (6 directions), 1000 (30 directions) and 2500 (30 directions) s / mm², TE/TR = 82 ms/9 s, acquisition matrix = 128 x 128, FOV = 256 x 256 mm², slice thickness = 2 mm, 72 slices, 1500 Hz/px readout bandwidth and total acquisition duration of 11.4 minutes.

Image processing

T₂ FLAIR images were co-registered to the T_{1w} images within the same imaging session using *SPM12* (UCL, UK). Diffusion-weighted images were motion corrected, masked and eddy current corrected using *FSL 6.0.1* (the University of Oxford, UK). FWF maps were created in native space using an in-house routine. In line with previous publications, only b-values < 2000 s / mm² were used.^{18,19,49} The resultant parametric maps from the diffusion imaging were subsequently also co-registered to the corresponding T_{1w} image.

MS brain lesions were outlined by *icobrain ms* (Icometrix, Belgium), an FDA-approved and CE-marked radiological services provider. Subcortical structures were segmented on the T_{1w} images using the *FSL FIRST* tool after lesion filling had been performed. The segmented structures included separate measures for both left and right hemispheres of the following subcortical structures: thalamus, caudate, putamen, pallidum, amygdala and accumbens. Measures over both hemispheres included whole white and grey matter, combined brain stem and fourth ventricle area and hippocampus. For analysis, the measures for left and right

hemispheres of the subcortical structures were averaged. In the whole white and grey matter analysis, areas with lesions were excluded by artificially extending each lesion with a $5 \times 5 \times 5 \text{ mm}^3$ Gauss filter and subtracting it from the analysis mask.

As well as computing total volume and volume changes over time, segmented structures were used for masking regions of interest (ROI) in the free water maps.

To account for partial volume effects in these ROI masks when estimating free water, their size was reduced by applying the erosion function *imerode* in *Matlab 9.5.0* (the MathWorks, Natick, MA). The strength of the erosion was set to be approximately $1/3$ of total volume before erosion.

For the general linear model (GLM) analysis, individual T_1w images were transformed into standard space based on the MNI152 T_1w template. The transformation was then applied to the FWF maps from the first visit and images were subsequently smoothed with a $3 \times 3 \times 3 \text{ mm}^3$ Gaussian filter.

Statistical analysis

Statistical analysis was performed in *Matlab 9.5.0* (the MathWorks, Natick, MA). Pearson correlations were computed between MRI data (total white and grey matter, subcortical structures volume, volume change between timepoints, FWF) and all test scores and responses on the questionnaires. Total number of test variables in the correlation matrix was 18 with 46 observations each. Resultant p-value matrices were corrected for multiple comparisons using false discovery rate (FDR) testing and only FDR-corrected p-values are reported.^{50,51} A correlation coefficient greater than $r = \pm 0.3$ with a P -value < 0.05 (FDR-corrected) was considered significant.

To investigate whether FWF at the first visit can predict HADS and the other test and questionnaire scores at the most recent clinical assessment (the third visit), structural equation modeling (SEM) was set up using a partial least squares algorithm (PLS-SEM).⁵² For this, the 18 measurement variables belonging to the FWF mean values of each subcortical region and individual test and questionnaire scores were grouped into the following five latent constructs: FWF in subcortical region (including thalamus, caudate, putamen, pallidum, amygdala and accumbens), EDSS, BICAMS (including SDMT, CVLT-II and BVMT-r), FSMC (including

cognitive, motor and total fatigue scores) and HADS (including an anxiety, depression and a total anxiety/depression score). The model was set up as a formative measurement model.

For the GLM analysis focusing on symptoms of depression at the last clinical visit, subjects were grouped by their HADS depression subscore 0-2 ($n = 23$) and 2-9 ($n = 23$) as recorded at the two-year follow-up and a two-sample t-test was performed to the FWF maps of the baseline visit in *SPM12* (UCL, London, UK) with age as a covariate. The cut-off value of HADS depression subscore was chosen as it provided equal sample sizes.

Data availability

The data that support the findings of this study are available on request from the corresponding author. The data are not publicly available due to containing information that could compromise the privacy of research participants. Approval for data sharing is subject to approval by the author’s local ethics committee and a formal data sharing agreement.

Results

Out of the 65 recruited subjects with relapsing-remitting MS, two opted out and withdrew their consent, two did not attend the baseline visit, four did not return after the baseline visit and seven did not attend for the two-year follow up. One subject missed the one-year follow-up but had completed the baseline and two-year follow-up visits, and was not removed from the dataset. A further three subjects had incomplete cognitive scores at the two-year follow-up visit and were therefore excluded. In total, 46 participants had complete data collection at baseline and two-year follow-up and were therefore included in this study, see Table 1 for participant demographics. MRI alongside with a neuropsychological examination of cognitive and emotional function at baseline as well as at two follow-up visits were scheduled at approximately one and two years after the baseline examination, respectively. The average time between MRI and neurocognitive assessment was 7.5 ± 31.0 days.

Table 1 Study participant demographics

Characteristic	Baseline	1-year follow-up	2-year follow-up	1-year follow-up HADS D < 2	2-year follow-up HADS D > 2
Number of participants:	46	45	46	23	23
Age (years):					

Median	37.9	38.8	39.9	36.9	41.9
Range	21.0-63.8	21.9-64.8	22.8-65.9	26.5-55.9	22.9-65.9
Gender:					
Female	32	31	32	17	15
Male	14	14	14	6	8
EDSS:	1.3 ± 0.9	1.5 ± 0.8	1.5 ± 0.9	1.3 ± 0.8	1.8 ± 0.9
FSMC:					
Motor	14.1 ± 9.7	14.8 ± 10.4	15.5 ± 10.7	9.9 ± 8.9	21.0 ± 9.7
Cognitive	14.2 ± 10.5	14.8 ± 10.8	14.9 ± 9.9	9.4 ± 8.5	20.4 ± 8.0
Total	28.3 ± 19.6	29.6 ± 20.9	30.4 ± 20.1	19.3 ± 16.8	41.5 ± 17.0
HADS:					
Anxiety	5.0 ± 3.8	4.3 ± 2.7	4.6 ± 3.2	2.9 ± 2.7	6.3 ± 3.0
Depression	2.7 ± 3.0	2.2 ± 2.2	2.7 ± 2.6	0.6 ± 0.6	4.7 ± 2.3
Total	7.7 ± 6.0	6.4 ± 4.1	7.2 ± 5.2	3.5 ± 3.0	11.0 ± 4.3
BICAMS:					
SDMT	56.1 ± 10.7	58.5 ± 12.0	58.8 ± 11.7	61.9 ± 9.8	55.8 ± 12.9
CVLT-II	55.4 ± 10.9	57.8 ± 8.7	58.2 ± 10.4	61.8 ± 9.5	54.6 ± 10.1
BVMT-r	27.4 ± 5.3	28.3 ± 5.95	27.3 ± 5.6	28.1 ± 4.7	26.4 ± 6.3

EDSS: Expanded disability status scale. FSMC = Fatigue scale for motor and cognitive functions, HADS = Hospital anxiety and depression scale, BICAMS = Brief international cognitive assessment for Multiple Sclerosis, SDMT = Symbol digit modalities test, CVLT-II = the learning trials of the 2nd edition of the California Verbal Learning Test, BVMT-r = Revised brief visuospatial memory test.

In the following, if not stated otherwise, all results are reported after multiple comparison testing corrections.

Mean values of FWF for each visit and ROI before and after ROI erosion are given in Supplementary Table 1.

Association of FWF in subcortical structures and depression and anxiety score (HADS)

FWF at baseline did not correlate with HADS scores at baseline or the one-year follow-up, however FWF at baseline was correlated with the combined (anxiety and depression) HADS scores at the two-year follow-up in the: Thalamus ($r = 0.40$, $P = 0.01$), putamen ($r = 0.32$, $P = 0.03$), pallidum ($r = 0.37$, $P = 0.01$), hippocampus ($r = 0.43$, $P = 0.004$), amygdala ($r = 0.41$, $P = 0.01$) and accumbens ($r = 0.40$, $P = 0.01$).

Association of FWF in subcortical structures and depression subscore

FWF at baseline was also correlated with the HADS depression subscore at the two-year follow-up in the: Thalamus ($r = 0.47$, $P = 0.002$), putamen ($r = 0.40$, $P = 0.01$), pallidum ($r =$

0.42, $P = 0.005$), hippocampus ($r = 0.42$, $P = 0.005$), amygdala ($r = 0.34$, $P = 0.02$) and accumbens ($r = 0.44$, $P = 0.003$).

Testing for a relationship between depression and FWF in subcortical structures using PLS-SEM and GLM analysis

The PLS-SEM model converged in 301 iterations and fulfilled construct reliability of all latent variables with a Cronbach Alpha > 0.7 . Predictive nature of baseline diffusion measures and the HADS depression subscore was confirmed in the SEM analysis (total effect size of path: 0.46).

Fig. 1 shows the results of the two-sample t-test GLM analysis. Clusters of high t-values on this map highlight significantly different ($P = 0.001$, uncorrected) areas in the FWF maps between participants with low (< 2 , $n = 23$) vs. higher (> 2 , $n = 23$) HADS depression subscore. Clusters are clearly discernable in the amygdala, hippocampus and thalamus. Additional clusters were also seen in the corpus callosum, cerebellar vermis and precuneus.

Figure 1 Resultant t-map ($P = 0.001$, uncorrected) of the GLM analysis of FWF maps at the baseline visit between subjects with HADS depression < 2 ($n = 23$) and HADS depression > 2 ($n = 23$) at the two-year follow-up visit overlaid on the sagittal MNI152 T₁-weighted template. Clusters highlight differences between the groups and are seen in the amygdala and hippocampus area (slices -20 to -10) on the left and in the magnification panel on the right as the top image. Clusters are also seen in the thalamus area (slices -6 to +6) and in the magnification panel on the right as the bottom image. Additional clusters were seen in

the corpus callosum (slices -10 to +16), cerebellar vermis area (slices -4 to +2) and precuneus (slices -20 to -10).

Association of FWF in subcortical structures and anxiety subscore

FWF at baseline was correlated with the HADS anxiety subscore at the two-year follow-up between the: Hippocampus ($r = 0.35$, $P = 0.02$) and amygdala ($r = 0.38$, $P = 0.01$).

In summary, FWF in all the segmented subcortical structures except for the caudate were correlated with one of the HADS metrics. FWF in the subcortical structures was also not correlated to any other clinical measurement at any timepoint.

Association of FWF in total white and grey matter and cognitive measures (BICAMS)

Diffusion microstructure in cortical grey matter was found to be negatively correlated with all BICAMS metrics at their respective timepoints: (r-BVMT - baseline visit: $r = -0.46$, $P < 0.001$, one-year follow-up: $r = -0.41$, $P < 0.001$), (SDMT- baseline visit $r = -0.58$, $P < 0.001$, one-year follow-up visit: $r = -0.54$, $P < 0.001$) the California Verbal Learning Test-II (CVLT-II – baseline visit: $r = -0.37$, $P = 0.001$, one-year follow-up visit: $r = 0.33$, $P = 0.004$).

Association of volumetric measures and cognition (BICAMS) and fatigue (FSMC)

Hippocampal and accumbens volumes at baseline were positively correlated with individual BICAMS metrics such as the SDMT ($r = 0.49$, $P < 0.01$) and the r-BVMT ($r = 0.35$, $P = 0.02$). This relationship was maintained in the one-year follow-up visit where the hippocampus volume again correlated with SDMT ($r = 0.37$, $P = 0.03$) and the accumbens volume with r-BVMT ($r = 0.36$, $P = 0.04$). At the two-year follow-up, SDMT remained moderately correlated with hippocampus volume ($r = 0.40$, $P = 0.02$) and accumbens volume ($r = 0.48$, $P = 0.003$).

At the two-year follow-up, thalamus volumes were correlated with cognitive fatigue ($r = -0.37$, $P = 0.03$) and combined fatigues score for motor and cognitive functions ($r = -0.35$, $P = 0.04$). The hippocampus ($r = -0.43$, $P = 0.01$) as well as the amygdala ($r = -0.37$, $P = 0.03$) were correlated with cognitive fatigue subscores. The hippocampus volume was also correlated with

the motor subscore from the fatigue scale ($r = -0.35$, $P = 0.04$) and the total fatigue scores ($r = -0.40$, $P = 0.02$).

Results from the volumetric analysis of the subcortical structures can be found in Supplementary Table 2 with atrophy measures in percent in Supplementary Tables 3 and 4. No statistically significant correlations between atrophy as a percentage-change and neuropsychological test scores were found.

Discussion

Changes in cognitive and motor function in MS are most generally recorded using a palette of different tests that aim to record different sets of performances. Such tests include the EDSS scale,³ 9-hole peg test,⁵³ timed 25-foot walk and/or more specialized tests focusing on the auditory and visual system.^{54,55} In addition, quality of life, mood and mental health^{56,57} may be recorded with additional questionnaires that aim to register symptoms such as fatigue, pain and depression.

Recommendations have been made on which assessment tools to include in monitoring of MS in standard practice and in clinical trials. The administration of these tools can, however, be time-consuming. Key targets of shortened exams, such as the BICAMS test,^{45,58} are to cover a broad-enough range of cognitive domains to be sensitive enough to monitor cognitive decline and changes in relation to treatment. We believe there is benefit in adding questionnaires such as the HADS and FSMC. The HADS has been widely-used and by using well-defined score cut-offs for depression (above 8 out of a possible 21 a.u.), a meta-analysis of 747 publications found a sensitivity and specificity of HADS of approximately 0.80 using the Cronbach alpha.⁵⁹

We observed significant correlations between free-water in the subcortical structures and the HADS depression and anxiety scores. As a means to correct for partial volume, we used erosion to shrink the ROIs, thereby minimizing the effect of neighboring structures and tissues to our results. The FWF diffusion metric for a given ROI increased after erosion, implying that our findings are not related to increased diffusion due to partial volume with cerebrospinal fluid in the extracellular space that may have increased due to atrophy. If partial volume and atrophy were indeed a driver, we would have expected decreased FWF values after erosion. We therefore suggest that our observation is instead due to changes within the subcortical

Commented [NP(5): Where there any correlations to the individual scores of the BICAMS sub scales (SDMT, BVL, CVLT) and if there were, then they should also be discussed.

structures that could be driven by grey matter changes in those areas. It would be beneficial to investigate for the presence of subcortical grey matter lesions; however identification of cortical and subcortical grey matter lesions in standard MR images is difficult.⁶⁰ It has been shown that cortical grey matter aberrations are present in the earliest stages of MS⁶¹ and a post mortem study did confirm that both focal demyelinating lesions and diffuse neurodegeneration are common in the deep grey matter of MS patients.⁶² The latter study also found that demyelination was most present in the hippocampus and caudate and could already be seen at an early disease stage.⁶² An investigation into demyelination and lesions in the deep grey matter would be of future interest using specialized MRI sequences such as double inversion recovery imaging (DIR) that can aid grey matter lesion detection.⁶⁰

We observed correlations between our free water diffusion metric and both the anxiety and depression subscores in nearly all subcortical structures. Hyperactivity of the hypothalamic-pituitary-adrenal axis has been suggested to be an endocrine basis for the development of depression^{24,63} and it is therefore not surprising to see the hippocampus and thalamus presenting with increased free-water, as seen in our study.

Only the caudate did not exhibit any correlation between the depression subscore and free water diffusion index. This is consistent with earlier research, suggesting that the caudate does not play a role in mood, depression or fatigue, but is rather associated with motor function and cognitive processes such as memory and learning.⁶⁴⁻⁶⁶

In our sample, HADS depression scores at baseline were 2.7 ± 3.0 out of possible 21 points and 5.0 ± 3.8 for anxiety, again out of possible 21 points and on the two-year follow-up 2.7 ± 2.6 and 2.9 ± 2.7 for depression and anxiety respectively. However, the theoretical upper maximum of 21 points is rarely scored, a cut-off of > 7 is usually used to define a subject as suffering clinically from depression or anxiety.⁵⁹ Our sample therefore represents a good range between mild symptoms and what would qualify as clinical depression. Our results suggest that FWF may be a sensitive enough technique to detect depression early.

We employed false discovery rate testing as a means to correct for multiple comparisons. We chose this method as it has been previously employed²⁸ in this type of study and since Bonferroni-Holm or permutations tests were deemed unsuitable for this dataset: While the Bonferroni-Holm method penalizes a large number of comparison variables, permutation testing is more suitable for larger datasets such as those from genome sequencing.⁶⁷ To test

whether baseline visit FWF could be used to predict HADS, we used a PLS-SEM analysis which overall confirmed the link between FWF and HADS. To move away from a ROI-based analysis we also employed whole brain GLM voxel-wise statistical testing that also highlighted statistically significant differences in only a few distinct regions with strong clusters in the amygdala, hippocampus and thalamus, similar to our ROI-based correlation analysis results. This also highlights that not only are there differences in the subcortical structures that predict depression symptoms, but also that these are varied and their strength can clearly differentiate the cohort into two groups. The GLM analysis results did unfortunately not survive FDR correction, possibly due to the reduction in sample size by splitting our cohort into two sub-groups. Aside from the clusters seen in the amygdala, hippocampus and thalamus, we also observed clusters in the corpus callosum, precuneus and cerebellum. While we attribute the cluster in the corpus callosum and precuneus possibly to artefacts, the strength and distinctiveness of the cluster in the cerebellum is striking and requires further investigation.

The individual metrics of the BICAMS test battery correlated with the free water diffusion index in the cortical grey matter. A recent study by Genç *et al.*⁶⁸ that employed a diffusion-based neurite density metric in 498 participants demonstrated a strong association between cortical grey matter and performance on an IQ test. This may imply that diffusion in the cortical grey matter may also influence the performance in the tasks included in the BICAMS test battery.

A recent cross-sectional study by Fleischer *et al.*²⁸ demonstrated that subcortical volumes could be used as an early predictor of fatigue in MS. Fleischer *et al.*²⁸ found mainly that the caudate volume at baseline correlated with fatigue at a follow-up visit 4 years later. In our study we saw correlations at the two-year follow-up between cognitive fatigue and the thalamus, hippocampus and amygdala. While we did not see a correlation between the caudate volume and fatigue, the highest correlation we observed was in the hippocampus ($r = -0.43$, $P = 0.01$), which the author of the Fleischer *et al.*²⁸ study suggested is involved mainly at the onset of MS. We therefore attribute the lack of caudate findings to the short disease duration in our cohort.

FWF is a diffusion microstructure parameter that can be computed in a few seconds from standard diffusion tensor imaging (DTI) acquisitions. Many centers include DTI acquisitions to facilitate fractional anisotropy (FA) and DTI tractography estimation, making all these studies potentially compatible with FWF estimation. More complex models that require longer,

dedicated acquisitions such as restricted spectrum imaging (RSI)⁶⁹ or neurite orientation dispersion and density imaging (NODDI)⁷⁰ also include an isotropic diffusion estimate which should provide similar information to FWF. It may therefore be possible to reproduce our results with one of these models.

Our data may suggest that processes related to MS in the subcortical structures may contribute to the development of depression symptoms. If early changes in FWF can predict onset and severity of depression, clinical decisions can be taken to prepare the patients and their families with complimentary treatment for depression such as cognitive behavioural therapy.

Acknowledgments

The authors would like to thank the MS patients for participation in the study, and express their thanks to the Health Authorities of Western Norway for funding of the project. We are grateful to Nuno Pedrosa de Barros from Icometrix for providing the lesion analyses and to scientists Hauke Bartsch and Erling Andersen, Haukeland

University Hospital, for their involvement in MRI data collection, handling and storage.

We are thankful to the team of neuroradiologists at the Department of Radiology, Haukeland University Hospital, for MR image reading and reporting. We are obliged to MS nurses Anne-Britt Rundhovde Skår and Randi C. Haugstad at the Norwegian MS Competence Center, Haukeland University Hospital, for their invaluable contribution to the study, and to health secretary Angunn Solberg at the outpatient clinic at the Department of Neurology, Haukeland University Hospital, for her contribution to logistics.

Funding

MMIV is jointly hosted by Haukeland University Hospital and University of Bergen and supported as a centre by grants from the Trond Mohn Foundation.

Neuro-SysMed is jointly hosted by Haukeland University Hospital and University of Bergen and supported as a centre for Clinical Treatment Research (FKB) by grants from The Research Council of Norway, project number 288164.

Ellen Skorve has received majority of funding through PhD-scholarship from the Health Authorities of Western Norway (3-year fellowship).

Additional financial support for the MRI investigation study was provided by Dr. Niels Vilhelm Henrichsen and wife Anna Henrichsens Legacy Fund.

Competing interests

Ellen Skorve has received initial funding for this study through an unrestricted research grant from Novartis (project planning and inclusion phase).

Øivind Torkildsen has received speaker honoraria from and served on scientific advisory boards for Biogen, Sanofi-Aventis, Merck and Novartis.

Kjell-Morten Myhr has received unrestricted research grants to his institution, scientific advisory board or speaker honoraria from Biogen, Merck, Novartis, Roche and Sanofi; and has participated in clinical trials organized by Biogen, Merck, Novartis, Roche and Sanofi.

All other authors report no competing interests.

Supplementary material

Average free water measures for the different subcortical areas before and after partial volume correction for all participants at the individual timepoints are given in Supplementary Table 1. Supplementary Table 2 summarises the mean volumes of the subcortical structures for all participants at the individual timepoints with percentage changes between timepoints given in Supplementary Tables 3 & 4.

Supplementary Table 1 Average FWF index of segmented subcortical nuclei in arbitrary units

Structure	Baseline (a.u.) before erosion	1-year follow-up (a.u.) before erosion	Baseline (a.u.) after erosion	1-year follow-up (a.u.) after erosion
Thalamus	0.24 ± 0.04	0.24 ± 0.05	0.26 ± 0.04	0.26 ± 0.05
Caudate	0.25 ± 0.05	0.25 ± 0.06	0.23 ± 0.05	0.24 ± 0.06
Putamen	0.13 ± 0.07	0.13 ± 0.09	0.22 ± 0.08	0.23 ± 0.08
Pallidum	0.13 ± 0.07	0.13 ± 0.09	0.25 ± 0.08	0.26 ± 0.08
Hippocampus	0.24 ± 0.03	0.24 ± 0.05	0.26 ± 0.04	0.26 ± 0.05
Amygdala	0.26 ± 0.06	0.26 ± 0.09	0.23 ± 0.08	0.23 ± 0.08
Accumbens	0.15 ± 0.05	0.16 ± 0.08	0.18 ± 0.08	0.19 ± 0.07
Whole white matter	0.25 ± 0.01	0.25 ± 0.01	N.A.	N.A.
Whole grey matter	0.30 ± 0.02	0.30 ± 0.02	N.A.	N.A.

No diffusion data was available for the 3-year follow-up.

Supplementary Table 2 Average volumes of segmented subcortical nuclei in mm³

Structure	Baseline (mm ³)	1-year follow-up (mm ³)	2-year follow-up (mm ³)
Thalamus	8493 ± 967.0	8449 ± 874.6	8331 ± 858.7
Caudate	3460 ± 459.6	3443 ± 430.4	3402 ± 404.2
Putamen	5044 ± 609.5	5021 ± 538.9	5012 ± 474.7
Pallidum	1865 ± 217.2	1845 ± 224.8	1845 ± 179.1
Hippocampus	3470 ± 523.5	3420 ± 496.4	3358 ± 455.7
Amygdala	1329 ± 207.7	1338 ± 208.7	1273 ± 220.3
Accumbens	496 ± 107.6	493 ± 98.4	484 ± 102.1

Supplementary Table 3: Volume changes of the deep gray matter nuclei between TP1 and TP2 in percent of total volume.

Thalamus	Caudate	Putamen	Pallidum	Hippocampus	Amygdala	Accumbens
0 ± 0.5%	0 ± 0.3%	0 ± 0.5%	-1 ± 0.75%	-1 ± 0.9%	+2 ± 1.3%	0 ± 1.0%

Supplementary Table 4 Volume changes of the deep gray matter nuclei between TP1 and TP3 in percent of total volume

Thalamus	Caudate	Putamen	Pallidum	Hippocampus	Amygdala	Accumbens
-2 ± 0.35%	-1 ± 0.2%	-1 ± 0.4%	0 ± 0.3%	-2 ± 1.9%	-4 ± 1.2%	-3 ± 0.9%

References

1. Thompson AJ, Baranzini SE, Geurts J, Hemmer B, Ciccarelli O. Multiple sclerosis. *Lancet (London, England)*. 2018;391(10130):1622-1636. doi:10.1016/S0140-6736(18)30481-1
2. Thompson AJ, Banwell BL, Barkhof F, et al. Diagnosis of multiple sclerosis: 2017 revisions of the McDonald criteria. *Lancet Neurol*. 2018;17(2):162-173. doi:10.1016/S1474-4422(17)30470-2
3. Kurtzke JF. Rating neurologic impairment in multiple sclerosis: an expanded disability status scale (EDSS). *Neurology*. 1983;33(11):1444-1452. doi:10.1212/wnl.33.11.1444
4. Grist JT, Riemer F, McLean MA, et al. Imaging intralesional heterogeneity of sodium concentration in multiple sclerosis: Initial evidence from ²³Na-MRI. *J Neurol Sci*. 2018;387. doi:10.1016/j.jns.2018.01.027
5. Ghione E, Bergsland N, Dwyer MG, et al. Aging and Brain Atrophy in Multiple Sclerosis. *J Neuroimaging*. 2019;29(4):527-535. doi:10.1111/jon.12625
6. B B, M C, F M, et al. Functional connectivity changes within specific networks

parallel the clinical evolution of multiple sclerosis. *Mult Scler.* 2014;20(8).
doi:10.1177/1352458513515082

7. Staffen W, Mair A, Zauner H, et al. Cognitive function and fMRI in patients with multiple sclerosis: evidence for compensatory cortical activation during an attention task. *Brain.* 2002;125(Pt 6):1275-1282. doi:10.1093/brain/awf125
8. Bisecco A, Rocca MA, Pagani E, et al. Connectivity-based parcellation of the thalamus in multiple sclerosis and its implications for cognitive impairment: A multicenter study. *Hum Brain Mapp.* 2015;36(7):2809-2825. doi:10.1002/hbm.22809
9. Alexander AL, Lee JE, Lazar M, Field AS. Diffusion tensor imaging of the brain. *Neurotherapeutics.* 2007;4(3):316-329. doi:10.1016/j.nurt.2007.05.011
10. Stejskal EO, Tanner JE. Spin Diffusion Measurements: Spin Echoes in the Presence of a Time-Dependent Field Gradient. *J Chem Phys.* 1965;42(1):288-292. doi:10.1063/1.1695690
11. Beaudoin A-M, Rheault F, Theaud G, et al. Modern Technology in Multi-Shell Diffusion MRI Reveals Diffuse White Matter Changes in Young Adults With Relapsing-Remitting Multiple Sclerosis. *Front Neurosci.* 2021;15:665017. doi:10.3389/fnins.2021.665017
12. Zhang J, Cortese R, De Stefano N, Giorgio A. Structural and Functional Connectivity Substrates of Cognitive Impairment in Multiple Sclerosis. *Front Neurol.* 2021;12:671894. doi:10.3389/fneur.2021.671894
13. Cercignani M, Gandini Wheeler-Kingshott C. From micro- to macro-structures in multiple sclerosis: what is the added value of diffusion imaging. *NMR Biomed.* 2019;32(4):e3888. doi:10.1002/nbm.3888
14. Lakhani DA, Schilling KG, Xu J, Bagnato F. Advanced Multicompartment Diffusion MRI Models and Their Application in Multiple Sclerosis. *AJNR Am J Neuroradiol.* 2020;41(5):751-757. doi:10.3174/ajnr.A6484
15. Pierpaoli C, Jones DK. Removing CSF Contamination in Brain DT-MRIs by Using a Two-Compartment Tensor Model. In: *Proceedings of the 12th Annual Meeting of ISMRM, Kyoto.* ; 2004.
16. Parker D, Ould Ismail AA, Wolf R, et al. Freewater estimator using interpolated initialization (FERNET): Characterizing peritumoral edema using clinically feasible diffusion MRI data. *PLoS One.* 2020;15(5):e0233645. doi:10.1371/journal.pone.0233645
17. Hoy AR, Koay CG, Kecskemeti SR, Alexander AL. Optimization of a free water

- elimination two-compartment model for diffusion tensor imaging. *Neuroimage*. 2014;103:323-333. doi:10.1016/j.neuroimage.2014.09.053
18. Pasternak O, Shenton ME, Westin C-F. Estimation of extracellular volume from regularized multi-shell diffusion MRI. *Med Image Comput Comput Assist Interv*. 2012;15(Pt 2):305-312. doi:10.1007/978-3-642-33418-4_38
 19. Starck L, Zaccagna F, Pasternak O, Gallagher FA, Grüner R, Riemer F. Effects of Multi-Shell Free Water Correction on Glioma Characterization. *Diagnostics*. 2021;11(12):2385. doi:10.3390/diagnostics11122385
 20. Wiig H. Pathophysiology of tissue fluid accumulation in inflammation. *J Physiol*. 2011;589(Pt 12):2945-2953. doi:10.1113/jphysiol.2011.206136
 21. Bullmore E. Inflamed depression. *Lancet*. 2018. doi:10.1016/S0140-6736(18)32356-0
 22. Patten SB, Marrie RA, Carta MG. Depression in multiple sclerosis. *Int Rev Psychiatry*. 2017;29(5):463-472. doi:10.1080/09540261.2017.1322555
 23. Feinstein A, Feinstein K. Depression associated with multiple sclerosis. *J Affect Disord*. 2001;66(2-3):193-198. doi:10.1016/S0165-0327(00)00298-6
 24. Feinstein A, Magalhaes S, Richard J-F, Audet B, Moore C. The link between multiple sclerosis and depression. *Nat Rev Neurol*. 2014;10(9):507-517. doi:10.1038/nrneurol.2014.139
 25. Uguz F, Akpınar Z, Ozkan I, Tokgoz S. Mood and anxiety disorders in patients with multiple sclerosis. *Int J Psychiatry Clin Pract*. 2008;12(1):19-24. doi:10.1080/13651500701330825
 26. Beiske AG, Svensson E, Sandanger I, et al. Depression and anxiety amongst multiple sclerosis patients. *Eur J Neurol*. 2008;15(3):239-245. doi:10.1111/j.1468-1331.2007.02041.x
 27. Eid K, Torkildsen ØF, Aarseth J, et al. Perinatal Depression and Anxiety in Women With Multiple Sclerosis: A Population-Based Cohort Study. *Neurology*. 2021;96(23):e2789-e2800. doi:10.1212/WNL.0000000000012062
 28. Fleischer V, Ciolac D, Gonzalez-Escamilla G, et al. Subcortical Volumes as Early Predictors of Fatigue in Multiple Sclerosis. *Ann Neurol*. January 2022. doi:10.1002/ana.26290
 29. Ousdal OT, Brancati GE, Kessler U, et al. The Neurobiological Effects of Electroconvulsive Therapy Studied Through Magnetic Resonance: What Have We Learned, and Where Do We Go? *Biol Psychiatry*. 2022;91(6):540-549. doi:10.1016/j.biopsych.2021.05.023

30. Ciccarelli O, Werring DJ, Wheeler-Kingshott CA, et al. Investigation of MS normal-appearing brain using diffusion tensor MRI with clinical correlations. *Neurology*. 2001;56(7):926-933. doi:10.1212/wnl.56.7.926
31. Cavallari M, Ceccarelli A, Wang G-Y, et al. Microstructural changes in the striatum and their impact on motor and neuropsychological performance in patients with multiple sclerosis. *PLoS One*. 2014;9(7):e101199. doi:10.1371/journal.pone.0101199
32. Hannoun S, Durand-Dubief F, Confavreux C, et al. Diffusion tensor-MRI evidence for extra-axonal neuronal degeneration in caudate and thalamic nuclei of patients with multiple sclerosis. *AJNR Am J Neuroradiol*. 2012;33(7):1363-1368. doi:10.3174/ajnr.A2983
33. Tovar-Moll F, Evangelou IE, Chiu AW, et al. Thalamic involvement and its impact on clinical disability in patients with multiple sclerosis: a diffusion tensor imaging study at 3T. *AJNR Am J Neuroradiol*. 2009;30(7):1380-1386. doi:10.3174/ajnr.A1564
34. Hasan KM, Halphen C, Kamali A, Nelson FM, Wolinsky JS, Narayana PA. Caudate nuclei volume, diffusion tensor metrics, and T(2) relaxation in healthy adults and relapsing-remitting multiple sclerosis patients: implications for understanding gray matter degeneration. *J Magn Reson Imaging*. 2009;29(1):70-77. doi:10.1002/jmri.21648
35. Amato MP, Ponziani G, Rossi F, Liedl CL, Stefanile C, Rossi L. Quality of life in multiple sclerosis: the impact of depression, fatigue and disability. *Mult Scler*. 2001;7(5):340-344. doi:10.1177/135245850100700511
36. Amato MP, Zipoli V, Portaccio E. Multiple sclerosis-related cognitive changes: a review of cross-sectional and longitudinal studies. *J Neurol Sci*. 2006;245(1-2):41-46. doi:10.1016/j.jns.2005.08.019
37. Bobholz JA, Rao SM. Cognitive dysfunction in multiple sclerosis: a review of recent developments. *Curr Opin Neurol*. 2003;16(3):283-288. doi:10.1097/01.wco.0000073928.19076.84
38. Chiaravalloti ND, DeLuca J. Cognitive impairment in multiple sclerosis. *Lancet Neurol*. 2008;7(12):1139-1151. doi:10.1016/S1474-4422(08)70259-X
39. Cortese M, Riise T, Bjørnevik K, et al. Preclinical disease activity in multiple sclerosis: A prospective study of cognitive performance prior to first symptom. *Ann Neurol*. 2016;80(4):616-624. doi:10.1002/ana.24769
40. Skorve E, Lundervold AJ, Torkildsen Ø, Myhr K-M. A two-year longitudinal follow-up of cognitive performance assessed by BICAMS in newly diagnosed patients with

- MS. *Mult Scler Relat Disord*. 2020;46:102577. doi:10.1016/j.msard.2020.102577
41. Skorve E, Lundervold AJ, Torkildsen Ø, Riemer F, Grüner R, Myhr K-M. Brief International Cognitive Assessment for MS (BICAMS) and global brain volumes in early stages of MS – a longitudinal correlation study. *Mult Scler Relat Disord*. November 2022;104398. doi:10.1016/J.MSARD.2022.104398
 42. Smith A. *Symbol Digit Modalities Test (SDMT). Manual (Revised)*. Los Angeles: Western Psychological Services; 1982.
 43. Delis DC, Kramer JH, Kaplan E, Ober BA. *California Verbal Learning Test—Second Edition*. San Antonio, TX: Psychological Corporation.; 2000.
 44. Benedict RHB. Brief visuospatial memory test - revised professional manual. *Psychol Assess Resour*. 1997;00.
 45. RH B, MP A, J B, et al. Brief International Cognitive Assessment for MS (BICAMS): international standards for validation. *BMC Neurol*. 2012;12. doi:10.1186/1471-2377-12-55
 46. Langdon DW, Amato MP, Boringa J, et al. Recommendations for a brief international cognitive assessment for multiple sclerosis (BICAMS). *Mult Scler J*. 2012;18(6). doi:10.1177/1352458511431076
 47. Penner IK, Raselli C, Stöcklin M, Opwis K, Kappos L, Calabrese P. The Fatigue Scale for Motor and Cognitive Functions (FSMC): validation of a new instrument to assess multiple sclerosis-related fatigue. *Mult Scler*. 2009;15(12):1509-1517. doi:10.1177/1352458509348519
 48. Zigmond AS, Snaith RP. The hospital anxiety and depression scale. *Acta Psychiatr Scand*. 1983;67(6):361-370. doi:10.1111/j.1600-0447.1983.tb09716.x
 49. Pasternak O, Sochen N, Gur Y, Intrator N, Assaf Y. Free water elimination and mapping from diffusion MRI. *Magn Reson Med*. 2009;62(3):717-730. doi:10.1002/mrm.22055
 50. Storey JD, Taylor JE, Siegmund D. Strong control, conservative point estimation and simultaneous conservative consistency of false discovery rates: A unified approach. *J R Stat Soc Ser B Stat Methodol*. 2004;66(1). doi:10.1111/j.1467-9868.2004.00439.x
 51. Storey JD. A direct approach to false discovery rates. *J R Stat Soc Ser B Stat Methodol*. 2002;64(3). doi:10.1111/1467-9868.00346
 52. Aria M. PLS-SEM Toolbox - File Exchange - MATLAB Central. 2022. https://se.mathworks.com/matlabcentral/fileexchange/54147-pls-sem-toolbox?s_tid=srchtitle. Accessed February 4, 2022.

53. Kellor M, Frost J, Silberberg N, Iversen I, Cummings R. Hand strength and dexterity. *undefined*. 1971. <https://www.semanticscholar.org/paper/Hand-strength-and-dexterity.-Kellor-Frost/04c062017818c824b7fa6dfbc3e343870d4001c3>. Accessed January 19, 2022.
54. Banos JH, Martin RC. California Verbal Learning Test-Second Edition. *Arch Clin Neuropsychol*. 2002;17(5):509-512. doi:10.1093/arclin/17.5.509
55. Benedict RHB, Schretlen D, Groninger L, Dobraski M, Shpritz B. Revision of the Brief Visuospatial Memory Test: Studies of normal performance, reliability, and validity. *Psychol Assess*. 1996;8(2):145-153. doi:10.1037/1040-3590.8.2.145
56. Kister I, Bacon TE, Chamot E, et al. Natural history of multiple sclerosis symptoms. *Int J MS Care*. 2013;15(3):146-158. doi:10.7224/1537-2073.2012-053
57. Yozbatiran N, Baskurt F, Baskurt Z, Ozakbas S, Idiman E. Motor assessment of upper extremity function and its relation with fatigue, cognitive function and quality of life in multiple sclerosis patients. *J Neurol Sci*. 2006;246(1-2):117-122. doi:10.1016/j.jns.2006.02.018
58. E S, AJ L, Ø T, KM M. The Norwegian translation of the brief international cognitive assessment for multiple sclerosis (BICAMS). *Mult Scler Relat Disord*. 2019;36. doi:10.1016/J.MSARD.2019.101408
59. Bjelland I, Dahl AA, Haug TT, Neckelmann D. The validity of the Hospital Anxiety and Depression Scale. An updated literature review. *J Psychosom Res*. 2002;52(2):69-77. doi:10.1016/s0022-3999(01)00296-3
60. Minagar A, Barnett MH, Benedict RHB, et al. The thalamus and multiple sclerosis: modern views on pathologic, imaging, and clinical aspects. *Neurology*. 2013;80(2):210-219. doi:10.1212/WNL.0b013e31827b910b
61. Pirko I, Lucchinetti CF, Sriram S, Bakshi R. Gray matter involvement in multiple sclerosis. *Neurology*. 2007;68(9):634-642. doi:10.1212/01.wnl.0000250267.85698.7a
62. Haider L, Simeonidou C, Steinberger G, et al. Multiple sclerosis deep grey matter: the relation between demyelination, neurodegeneration, inflammation and iron. *J Neurol Neurosurg Psychiatry*. 2014;85(12):1386-1395. doi:10.1136/jnnp-2014-307712
63. Bjedov B, Vidrih B, Tudor KI. *ANXIETY AND DEPRESSION AS COMORBIDITIES OF MULTIPLE SCLEROSIS*. Vol 33.; 2021. https://www.psychiatria-danubina.com/UserDocsImages/pdf/dnb_vol33_noSuppl 4/dnb_vol33_noSuppl 4_480.pdf. Accessed February 25, 2022.
64. Postle BR, D'Esposito M. Dissociation of human caudate nucleus activity in spatial

- and nonspatial working memory: an event-related fMRI study. *Cogn Brain Res.* 1999;8(2):107-115. doi:10.1016/S0926-6410(99)00010-5
65. White NM. Some highlights of research on the effects of caudate nucleus lesions over the past 200 years. *Behav Brain Res.* 2009;199(1):3-23. doi:10.1016/J.BBR.2008.12.003
66. Villablanca JR. Why do we have a caudate nucleus? *Acta Neurobiol Exp (Wars).* 2010;70(1):95-105. <http://www.ncbi.nlm.nih.gov/pubmed/20407491>. Accessed February 25, 2022.
67. Camargo A, Azuaje F, Wang H, Zheng H. Permutation – based statistical tests for multiple hypotheses. *Source Code Biol Med.* 2008;3(1):15. doi:10.1186/1751-0473-3-15
68. Genç E, Fraenz C, Schlüter C, et al. Diffusion markers of dendritic density and arborization in gray matter predict differences in intelligence. *Nat Commun.* 2018;9(1):1905. doi:10.1038/s41467-018-04268-8
69. White NS, Leergaard TB, D’Arceuil H, Bjaalie JG, Dale AM. Probing tissue microstructure with restriction spectrum imaging: Histological and theoretical validation. *Hum Brain Mapp.* 2013;34(2). doi:10.1002/hbm.21454
70. Zhang H, Schneider T, Wheeler-Kingshott CA, Alexander DC. {NODDI}: practical in vivo neurite orientation dispersion and density imaging of the human brain. *Neuroimage.* 2012;61(4):1000-1016. doi:10.1016/j.neuroimage.2012.03.072

Reviewer #1 (Remarks to the Author):

In this manuscript the authors assess links between early microstructural changes in subcortical volumes & depressive symptoms in a longitudinal study of 46 people with relapsing-remitting MS. Free water fraction was used to assess tissue microstructure, and a binary split was used to classify patients as having 'high' or 'low' depression scores. They find that all subcortical diffusion measures, except caudate, correlated with depression at 2 year visit. Correlations were carried out with depression, anxiety and a combined depression +anxiety score. The authors conclude that higher levels of free water in subcortical structures is linked to depression early in the course of MS (but not cognitive processes).

The study is interesting and the methods have been well carried out. Recruiting and retaining participants for these longitudinal studies is challenging – so the authors should be commended. There are some issues about the choice of outcome variables and whether the introduction perhaps misses out other studies of free water diffusion in MS. But otherwise the study reads well and is useful.

1. *In the introduction the authors state that free water fraction has previously been used in cancer imaging, but not in MS. However axonal water fraction can be derived from diffusion kurtosis imaging models – and has been used a few times in people with MS (see Hori et al., 2022 Mag Res in Med Sci). It might be worth mentioning these other approaches here.*

Response: Thank you for your kind comments and for highlighting the review by Hori et al. It is correct that the free water can be estimated from other diffusion methods as well. We have clarified this in the introduction and cited the suggested article on page 5 (insertions are marked in blue):

“It has previously been used to estimate the fraction of extracellular water in applications such as quantifying the contribution of edema in tumors and the tissue surrounding lesions in MS.¹⁶⁻²⁰ ~~but not in MS.~~ The FWF can also be estimated from other diffusion models.²¹”

²⁰Bergsland N, Dwyer MG, Jakimovski D, Weinstock-Guttman B, Zivadinov R. Diffusion tensor imaging reveals greater microstructure damage in lesional tissue that shrinks into cerebrospinal fluid in multiple sclerosis. *J Neuroimaging*. 2021;31(5):995-1002. doi:10.1111/jon.12891

²¹Hori M, Maekawa T, Kamiya K, et al. Advanced Diffusion MR Imaging for Multiple Sclerosis in the Brain and Spinal Cord. *Magn Reson Med Sci*. 2022;21(1). doi:10.2463/mrms.rev.2021-0091

2. *Also in the intro, the authors start mentioning links between the BICAMS and grey matter volumes in MS – but this trails off without noting what correlations were found. Can they present more details here?*

Response: We have changed this paragraph and included more details, see insertions in the text below (page 6):

“It has recently been shown in studies including the Brief International Cognitive Assessment for MS (BICAMS) tests,⁴³ ~~that the results on subsets from this test battery~~ psychometric tests of processing speed as well as verbal and visual memory were correlated with whole brain and grey matter volumes ~~measures~~ derived from an MRI examination at baseline.⁴⁴ After two years, the

authors found significant changes in global volumes that allowed differentiation of patients that were defined as either cognitively impaired or preserved.”

3. *Brain lesions were automatically segmented but it's not clear whether lesion-filling was carried out before estimating volumes?*

Response: Thank you for highlighting this. We have mentioned lesion-filling in the methods but it is not clear for which part of the analysis this was used. We have therefore changed the following paragraph on pages 8 and 9:

“Subcortical structures were segmented on the T₁w images using the *FSL FIRST* tool after lesion filling had been performed. (...)”

In the whole white and grey matter FWF analysis, areas with lesions were excluded by artificially extending each lesion with a 5 x 5 x 5 mm³ Gauss filter and subtracting it from the analysis mask. For the whole white and grey matter volume analysis, lesions were filled-in before measures were computed.”

4. *Do the authors really believe it is useful to combine Anxiety and Depression scores? Given that both have separate anatomical bases it's not clear what's gained by using this summed measure as an outcome.*

Response: We agree that measures of anxiety and depression should be handled as separate subscales, as we have done in most of our analysis. The particular test we used (HADS) includes questionnaires for both anxiety and depression and it is common for this test also to quote the combined score for both subtests. We have therefore included both individually as well as combined scores. We have made the differentiation between combined score and subscore more clear throughout the document.

5. *The authors test for association between free water diffusion in the whole grey matter and cognitive performance – but not with free water diffusion in individual grey matter structures. Was this because no associations were found? If so, it would be helpful to report this and consider in the discussion.*

Response: We only found associations between free water diffusion in the *cortical* grey matter and cognitive performance, but not in the sub-cortical or whole grey matter. We have clarified this on page 13:

“Association of FWF in total white and grey matter and cognitive measures (BICAMS)

Diffusion microstructure in cortical grey matter was found to be negatively correlated with all BICAMS metrics at their respective timepoints: (r-BVMT - baseline visit: $r = -0.46$, $P < 0.001$, one-year follow-up: $r = -0.41$, $P < 0.001$), (SDMT- baseline visit $r = -0.58$, $P < 0.001$, one-year follow-up visit: $r = -0.54$, $P < 0.001$) the California Verbal Learning Test-II (CVLT-II – baseline visit: $r = -0.37$, P

= 0.001, one-year follow-up visit: $r = 0.33$, $P = 0.004$). Diffusion microstructure in the subcortical structures and whole grey matter was not found to be correlated with any of the BICAMS test metrics.”

Reviewer #2 (Remarks to the Author):

What are the major claims of the paper? Are they novel and will they be of interest to others in the community and the wider field?

This paper will definitely be vital to MS research and the wider community. In a nutshell, the paper seeks to characterize neuroimaging correlates of affective and cognitive presentations in MS. Considering the high prevalence of some of these disorders in MS and the poorer outcomes and prognosis from existing co-morbidity, such research is vital not only for better understanding of etiopathology (of both MS and affective disorders) but also could be important in the identification of therapeutic end points during large scale clinical trials. Indeed, when MS occurs along with some of these presentations it is difficult to treat.

Response: Thank you for this kind comment.

1. *In addition to free water fraction (FWF), perhaps also considering other diffusion parameters such MD, FA, RD, AD, as well as NODDI parameters (ODI, ICVF, and ISOVF) may have made for stronger conclusions but that would also mean modifying methodology.*

Response: We agree with this statement and in particular, that metrics such as ISOVF would likely demonstrate the same results. We have emphasized this in the discussion as below, but do not want to add these other metrics with more results on top as we believe this could become quite long (as in addition to the volumetric measures) and potentially deflect from the main message.

Page.16:

More complex models that require longer, dedicated acquisitions such as restricted spectrum imaging (RSI)⁷⁶ or neurite orientation dispersion and density imaging (NODDI)⁷⁷ also include an isotropic diffusion estimate which should provide similar information to FWF. It may therefore be possible to reproduce our results with one of these models.

2. *Further reviews attached (in Word document)*

Response: Thank you for the additional comments in the Word document, we have addressed these as follows:

- Relapsing remitting MS has been abbreviated with RRMS.
- Sentences marked with reword have been reworded.
- Typos and language have been corrected as suggested.

More detail on individual comments below:

3. *Introduction: Similarly, cognitive function is also shown to be affected in patients with MS, at any stage and subtype of the disease.³⁵⁻³⁹ (please confirm this; may be more prevalent in PPMS-ref).*

Response: Thank you for this comment. Cognitive function impairment is indeed more prevalent in PPMS (and SPMS). We have added to p. 6 (insertions marked in blue):

“Similarly, cognitive function is also shown to be affected in patients with MS, at any stage and subtype of the disease.³⁷⁻⁴¹ The prevalence and severity of cognitive impairment appears greatest in PPMS and SPMS.⁴²”

4. *Introduce diffusion MRI in earlier paragraphs, include prevalence values for both cognitive impairment and depression in MS and explore literature that includes phenotypic variations. Add some phenotypic /clinical differences between RRMS and PPMS*

Response: Thank you for your comments. We have addressed these as follows: Diffusion MRI is introduced in the first paragraph on page 5. We have added more information on the difference between RRMS and PPMS to the introduction (p.4) which now reads:

“Most patients (85-90%) experience a relapsing-remitting disease course (RRMS), and 10-15% experience a primary progressive (PPMS) or secondary progressive (SPMS) course. A hallmark of RRMS is high disease activity in terms of lesions with distinct attacks and remission periods, while PPMS and SPMS have fewer lesions and are marked by a gradual worsening of symptoms without any distinct attacks or remission periods.¹”

More has been added on cognitive impairment and phenotypes to the paragraph on page 6:

“Similarly, cognitive function is also shown to be affected in patients with MS, at any stage and subtype of the disease.³⁷⁻⁴¹ The prevalence and severity of cognitive impairment appears greatest in PPMS and SPMS.⁴² It has recently been shown in studies including the Brief International Cognitive Assessment for MS (BICAMS) tests,⁴³ that the results on psychometric tests of processing speed as well as verbal and visual memory were correlated with whole brain and grey matter volumes measures derived from an MRI examination at baseline.⁴⁴ After two years, the authors found significant changes in these global volumes that allowed differentiation of patients that were defined as either cognitively impaired or preserved.

Generally, the prevalence of cognitive impairment in MS ranges from 34 to 65 % and depression is a symptom in one of four patients between the ages of 18-45.^{26,45} Zabad et al. found that patients suffering from PPMS were less at risk to suffer from major depression than RRMS patients.⁴⁶”

5. *Page 7 Methods: 46 participants were mentioned in Abstract.*

Response: 65 patients were recruited of which only 46 completed the full protocol that was used for this study. In line with reporting guidelines, we have opted to state the full number of

recruited subjects in the methods with details on dropouts on total numbers in the results section.

6. *Page 7 Methods: Not sure I understand this statement.*

Response: Thank you for highlighting this. We have simplified and re-phrased the sentence as follows:

“Diffusion weighted imaging (DWI) was performed as part of the MRI protocol at baseline and at the first follow-up, but not at the second follow-up.”

7. *Where there any correlations to the individual scores of the BICAMS sub scales (SDMT, BVL, CVLT) and if there were, then they should also be discussed.*

Response: We have updated the paragraph as follows and made the following insertions in the discussion on p. 16:

“All of the individual metrics of the BICAMS test battery correlated with the free water diffusion index in the cortical grey matter at the respective timepoints. A recent study by Genç *et al.*⁷³ that employed a diffusion-based neurite density metric in 498 participants, demonstrated a strong association between cortical grey matter and performance on an IQ test. This may imply that diffusion in the cortical grey matter may also influence the performance of the tasks included in the BICAMS test battery.

In addition to the diffusion findings, we found a significant correlation between the results on the SDMT and hippocampus volume at each of the respective timepoints and the r-BVMT to be correlated with the accumbens volume at both baseline and one-year follow up. The role of the hippocampus has been linked to memory, inhibition and spatial cognition and has been related to cognitive changes in diseases like Huntington’s and Alzheimer’s disease.⁷⁴ Our findings may be explained by characteristics of the SDMT: In the SDMT task, the examinee is asked to match numbers with geometrical shapes during a period of 90 seconds, and by this putting a strong load on both memory function and spatial cognition. The r-BVMT is characterized by memory and learning components.^{49,59} The correlation between r-BVMT and accumbens may therefore be attributed to the accumbens’ contribution to learning, memory and reward mechanisms, as was recently demonstrated in a model of dopaminergic neuron loss.⁷⁵

REVIEWERS' COMMENTS:

Reviewer #1 (Remarks to the Author):

The authors have addressed my concerns - and the manuscript is now improved. No further issues.

Reviewer #2 (Remarks to the Author):

Hello;

I am happy with the revised manuscript. My concerns have been addressed.

Good luck, and best wishes to the authors.